# SPILLED ENERGY IN LARGE LANGUAGE MODELS

**Adrian R. Minut** [1,2]    **Hazem Dewidar** [1,2]    **Iacopo Masi** [1]

🍥 Sapienza University of Rome, Italy   [1]🧘 OmnAI Lab [2]🐍 GLADIA

## ABSTRACT

We reinterpret the final Large Language Model (LLM) softmax classifier as an Energy-Based Model (EBM), decomposing the sequence-to-sequence probability chain into multiple interacting EBMs at inference. This principled approach allows us to track "energy spills" during decoding, which we empirically show correlate with factual errors, biases, and failures. Similar to Orgad et al. (2025), our method localizes the exact answer token and subsequently tests for hallucinations. Crucially, however, we achieve this without requiring trained probe classifiers or activation ablations. Instead, we introduce two completely training-free metrics derived directly from output logits: **spilled energy**, which captures the discrepancy between energy values across consecutive generation steps that should theoretically match, and **marginalized energy**, which is measurable at a single step. Evaluated on nine benchmarks across state-of-the-art LLMs (including LLaMA, Mistral, and Gemma) and on synthetic algebraic operations (Qwen3), our approach demonstrates robust, competitive hallucination detection and cross-task generalization. Notably, these results hold for both pretrained and instruction-tuned variants without introducing any training overhead. Code available at github.com/OmnAI-Lab/spilled-energy/

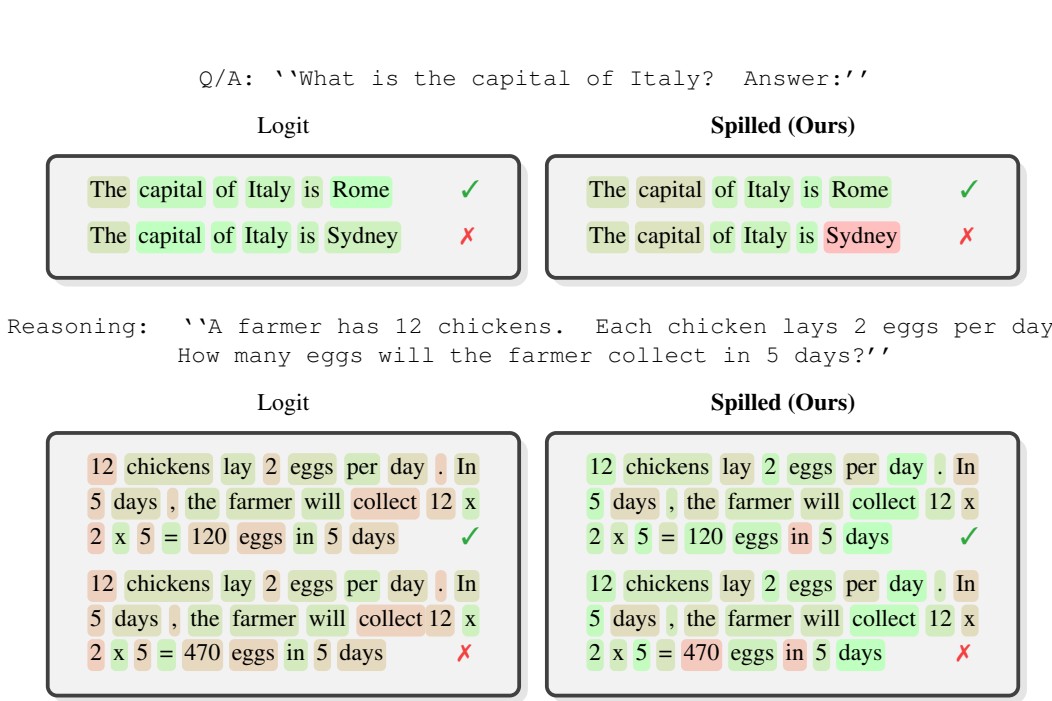

Figure 1: Color-coded comparison of hallucination detection with LLaMa-3 8B using logit confidence and **spilled energy**. Our method generalizes well across topics (e.g., Q&A, reasoning) and diverse LLMs. ✓ indicates a correct answer and ✗ an incorrect one. While **our approach focuses on the exact answer tokens** (e.g. Rome/Sydney and 120/470, see Section 4.2), here we apply min–max normalization to the full answer for visualization, as truthful ▬▬▬▬ hallucination.

## 1 INTRODUCTION

The widespread adoption of Large Language Models (LLMs) across various domains has brought increasing attention to their critical limitation: their tendency to generate incorrect or misleading information—commonly referred to as "hallucinations." This issue supports the idea that LLMs are just stochastic parrots (Bender et al., 2021) answering in a way that is statistically plausible with respect to the input prompt despite not having a real understanding of it. On the other side, recent reasoning capabilities proper to ChatGPT 4o (OpenAI-Team, 2023) or Deepseek (Liu et al., 2024) offer counter evidence to actually support this.

Ongoing research seeks to characterize and categorize hallucinations, setting them apart from other error types (Liu et al., 2022; Ji et al., 2023; Huang et al., 2023b; Rawte et al., 2023). At the same time, recent discussions have introduced terms such as confabulations (Millidge, 2023) and fabrications (McGowan et al., 2023), sometimes attributing a form of "intention" to LLMs—though the very idea of LLM "intentionality" and other human-like qualities remains contested (Salles et al., 2020; Serapio-García et al., 2023; Harnad, 2024). Research on LLM hallucinations can be categorized into two main branches: the first one is the extrinsic branch, where the hallucinations are measured with respect to the interpretation that humans give to those errors (Bang et al., 2023; Ji et al., 2023; Huang et al., 2023b; Rawte et al., 2023). The second branch was started by Kadavath et al. (2022b), proposing to study the hallucinations *within* the model itself. Following Kadavath et al. (2022b), the work in Li et al. (2024) proposes Inference-Time Intervention (ITI) as a way to improve the "truthfulness" of LLMs at inference time. ITI functions by altering model activations at inference time, steering them along specific directions within a restricted set of attention heads. Our work is also different from Yin et al. (2023), since we care about detecting errors in LLMs, whereas they introduce an automated methodology to detect when LLMs are aware that they do not know how to answer.

In this work, we follow the definition of hallucinations given by Orgad et al. (2025) as any form of error produced by an LLM—including factual mistakes, biased outputs, breakdowns in common-sense reasoning, and related issues. Like them, we also confirm that the truthfulness signal is concentrated in the "exact answer tokens." Nevertheless, unlike them, we abandon the idea of using a probe classifier (Belinkov, 2022) trained for each task and dataset. Given that LLMs are foundational models, user interactions typically occur *in the wild*, making it difficult to predict which probe classifier is best suited for detecting hallucinations in real-world scenarios. Furthermore, in this setting, classifier weights should not only be updated dynamically for each task, but the optimal token–layer combination is also dataset-dependent, which conflicts with the broad LLM applicability. Indeed, in the work by Orgad et al. (2025), the authors report:

> "We find that probing classifiers do not generalize across different tasks."

In our paper, we propose to solve this problem with a training-free method that generalizes better across different tasks and is mathematically principled using the framework of Energy-based Models (EBMs). Fig. 1 reports a qualitative comparison across tasks, comparing to the logit confidence. Additional samples are shown in Appendix D.2.

We reinterpret the final softmax classifier over the vocabulary of LLM as an EBM, taking inspiration from what Grathwohl et al. (2020) did for classifiers. This perspective enables us to decompose the sequence-to-sequence probability chain into multiple interacting EBMs that operate jointly during inference. Through this decomposition, we introduce the notion of "spilled energy" in LLM decoding and show empirically that such spill strongly correlates with errors. Given that our method is solely based on the mathematics of EBMs and the chain rule of probability, we do not have to train or tune our detector, striking a good generalization across tasks and LLMs. Building on this foundation, our contributions are as follows:

◇ Training-free, LLM hallucination detection generalizing across tasks using the EBM framework. We introduce a method for detecting hallucinations that requires no additional training, in contrast to prior work that relies on trained classifiers and ablations of model activations. Our approach directly reads values inside the LLM, enabling natural generalization across tasks and performing better than logit-based detection.

◇ Two energy-based metrics. We define two complementary measures of energy spills: (i) delta energy $\Delta E_{\boldsymbol{\theta}}(\mathbf{x}_{i:1})$, which captures discrepancies between energy values across two time steps that

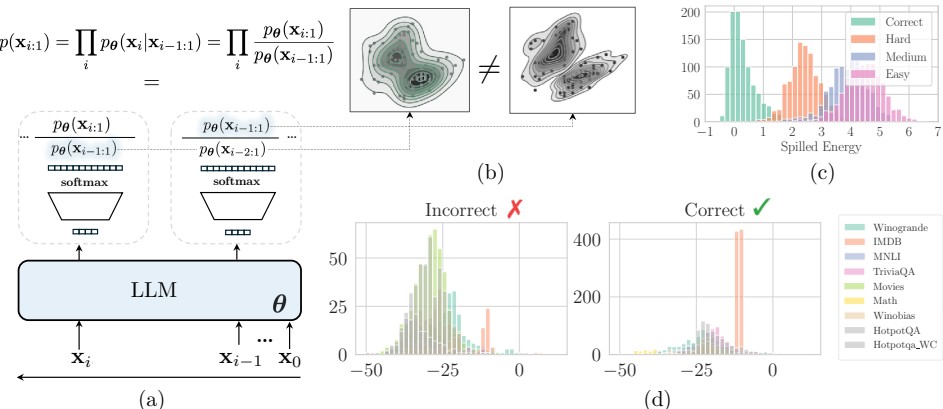

Figure 2: **How energy spills in LLMs**. (a) Language Modeling $p(\mathbf{x}_{i:1})$ is attained as a decomposition problem following the chain rule of probability, implemented as autoregressive: we recursively apply a discriminative classifier over the vocabulary $\mathcal{V}$ to attain generative modeling with larger context size i.e. $p(\mathbf{x}_i|\mathbf{x}_{i-1:1})$. (b) We reinterpret each discriminative classifier as a generative EBM, finding a connection between two quantities that should be the same across time steps yet are different. We call this difference "the spilled energy" $\Delta E_{\boldsymbol{\theta}}(\mathbf{x}_{i:1})$ in Eq. (8). (c) Given that we simply read values inside the LLM, our approach is training-free and correlates well with hallucinations on a synthetic math dataset with increasing difficulty; (d) histograms of spilled energy values, for incorrect and correct answers on all nine datasets using $\min$ pooling for Llama-3-Instruct. The two distributions are easily separable by using a simple threshold, resulting in a generalization across real-world tasks.

should be mathematically equivalent, and (ii) marginal energy $E_{\boldsymbol{\theta}}^m(\mathbf{x}_{i:1})$, which can be evaluated at a single time step.

◇ Scalable and generalizable analysis. Our framework is mathematically principled, training-free, and exhibits strong cross-dataset generalization. We scale our analysis to state-of-the-art LLMs, including Llama 3-8B-Instruct and Mistral-7B-Instruct, and demonstrate competitive performance across nine benchmarks, showing robustness across datasets and architectures.

Fig. 2(a) illustrates the core idea of our method: rather than using a naïve approach, such as simply recording the logit or training a probe classifier at the activations of the answer token, we first reinterpret the LLM as an autoregressive EBM via the chain rule of probabilities. We then further decompose each conditional probability, incorporating insights from Grathwohl et al. (2020). At the time step of the exact token $i - 1$, we extract the energy, which corresponds to the logit, and compare it with the marginal energy at the next time step $i$, corresponding to the denominator of the softmax. According to the chain rule, these two quantities should be identical; however, they differ in the LLM implementation—Fig. 2(b). We find that the discrepancy, which we term spilled energy $\Delta E_{\boldsymbol{\theta}}(\mathbf{x}_{i:1})$, correlates strongly with instances where the LLM produces an incorrect output—see Fig. 2(c). Moreover, its detection signal separates well correct and incorrect classes across datasets, reflecting the model's confidence, as shown in Fig. 2(d).

## 2 RELATED WORK

**EBM applications to Trustworthy AI.** EBMs have been applied to improve the reliability and interpretability of Deep Nets. For example, Energy-Based Out-of-Distribution Detection (OOD) (Liu et al., 2020) uses the energy score as a more robust alternative to softmax confidence. At the same time, Grathwohl et al. (2020) presents how to reinterpret a discriminative classifier as EBM to train models that are both discriminative and generative. Following this work, Zhu et al. (2021) provides new insights into the role of energy when training EBMs and robust classifiers using adversarial training. Instead, Mirza et al. (2024; 2025) explain adversarial attacks by reinterpreting the softmax classifier as an EBM, showing that these perturbations correspond to shifts in the underlying energy landscape.

**Foundations of Hallucination in LLMs.** LLMs are prone to diverse errors—including bias, reasoning failures, and generation of factually incorrect information unsupported by reliable sources. Karpowicz (2025) frames hallucination and imagination as mathematically identical phenomena, both emerging from a necessary violation of information conservation. Also Xu et al. (2025) provides a formal learning-theoretic proof that hallucinations are unavoidable. They define a *formal world* in which both the LLM and the ground-truth are computable functions, showing through classic results in computability theory, that no LLM can learn all such functions. As a consequence, hallucination is not just a practical artifact but a fundamental limitation of LLMs, valid even under idealized conditions. Recently, Kalai et al. (2025) showed that hallucinations come from the statistical problem of the pretraining methodology: minimizing the cross entropy naturally causes errors because it does not train the model to express uncertainty and say "I do not know." Kalai et al. (2025) proposes changing the evaluation practices to not reward models for guessing, but rather to mimic the human exams that penalize only wrong answers.

**Detecting and Mitigating LLM Hallucinations.** Orgad et al. (2025) train classifiers on the internal representations of the LLMs to predict, based on the features, the correctness of the answer. Given an LLM in a white-box setting, an input prompt, and the generated response $\hat{y}$, the classifier's task is to predict whether $\hat{y}$ is a hallucination. Orgad et al. suggested that LLMs may encode more factual knowledge in their latent subspaces than is revealed in their outputs. Gekhman et al. (2025) proposed a framework for studying hidden knowledge. Finally, Santilli et al. (2025) point out that uncertainty quantification in language models is often evaluated using metrics like AuROC. This shares biases between detection methods and correctness functions (e.g., length effects) that systematically distort results. One way to mitigate hallucinations is to act at the decoding stage, where the output generation can be steered Subramani et al. (2022). Steering vectors provide a straightforward way to control a model by adding a fixed vector to its activations (Dunefsky & Cohan, 2025). Fu et al. (2025) introduced DeepConf, a test-time method that leverages model-internal confidence signals to filter out low-quality reasoning traces during or after generation. Kuhn et al. (2023b); Fadeeva et al. (2024); Farquhar et al. (2024), and its follow-up by Kossen et al. (2025) in which they approximate the semantic entropy in a more efficient way. Constrained decoding approaches Li et al. (2023); Peng et al. (2023) modify token selection policies. Similarly, reinforcement learning with fact-based rewards Ouyang et al. (2022) has been used to bias decoding trajectories toward verifiable outcomes. Incorrect answers may also be given due to an ambiguous prompt: Kuhn et al. (2023a)'s CLAM framework uses few-shot prompts to classify a question's ambiguity and then asks the user to clarify.

## 3 Background and Foundations

### 3.1 Energy-Based Models

We give an overview of Energy-based Models (EBMs) and their use in discriminative classifiers.

**EBMs.** Energy-Based Models are a class of probabilistic models in which the probability distribution over data points $\mathbf{x}$ is defined in terms of an energy function $E_{\boldsymbol{\theta}}(\mathbf{x})$. The energy function, parameterized by a neural network $\boldsymbol{\theta}$ (Lecun et al., 2006), assigns a scalar energy to each configuration of $\mathbf{x}$, where lower energy values correspond to higher likelihood. The resulting probability distribution is given by $p_{\boldsymbol{\theta}}(\mathbf{x}) = \frac{\exp(-E_{\boldsymbol{\theta}}(\mathbf{x}))}{Z_{\boldsymbol{\theta}}}$ where $Z_{\boldsymbol{\theta}}$ denotes the partition function (normalizing constant), defined as $Z_{\boldsymbol{\theta}} = \sum_{\mathbf{x}} \exp(-E_{\boldsymbol{\theta}}(\mathbf{x}))$ for discrete $\mathbf{x}$, or equivalently $Z_{\boldsymbol{\theta}} = \int \exp(-E_{\boldsymbol{\theta}}(\mathbf{x})) \, d\mathbf{x}$ for continuous $\mathbf{x}$. Standard neural networks are often deterministic function approximators, mapping $\mathbf{x} \mapsto y$, EBMs instead define a full probability distribution over data or latent variables.

One of the strengths of EBMs is their flexibility in modeling arbitrary distributions without being tied to a specific parametric form. This flexibility comes from the fact that the energy function $E(\mathbf{x})$ can be defined in various ways. Training involves learning the parameters of the energy function such that the probability distribution $p_{\boldsymbol{\theta}}(\mathbf{x})$ matches the empirical distribution of the data. This is typically achieved using techniques like contrastive divergence, score matching, or maximum likelihood.

**Notation.** Let $\mathcal{V}$ denote the vocabulary of an LLM, i.e., the set of all tokens that can be processed as input and generated at each decoding step, with size $|\mathcal{V}| = V$. We shorten the sequence of tokens $\{\mathbf{x}_N, \ldots, \mathbf{x}_1\}$ as $\mathcal{X} = \{\mathbf{x}_{N:1}\}$, and $\mathbf{x}_i \in \mathcal{V}$ denotes the token in the $i$-th position along the sequence. We model the LLM as a function $\boldsymbol{\theta} : \mathbb{R}^{N \times V} \to \mathbb{R}^V$, implemented by a transformer, or any other sequence-to-sequence mechanism. For a sequence $\{\mathbf{x}_{i:1}\}$ as input, we write $\boldsymbol{\theta}(\mathbf{x}_{i:1})[k]$ to denote the

predicted logit assigned to the $k$-th token class in $\mathcal{V}$ for the $i + 1$ token in the sequence, as is standard in autoregressive LLM training (Ouyang et al., 2022).

## 3.2 AUTOREGRESSIVE LARGE LANGUAGE MODELS

Generative modeling has been pursued through a variety of approaches beyond autoregression (AR). Variational Autoencoders (VAEs) (Kingma & Welling, 2014) learn a probabilistic latent variable model by encoding inputs into a latent space and decoding samples back to the data domain. Generative Adversarial Networks (GANs) (Goodfellow et al., 2014) frame generation as a min-max game between a generator and a discriminator. The diffusion process has been incorporated into neural nets (Sohl-Dickstein et al., 2015) and, more recently, Diffusion Models (Ho et al., 2020) have emerged as a powerful class of generative models. While these paradigms differ in how they approximate the data distribution, AR models are special in their kind and take a more direct route by factorizing the joint probability of sequences into conditionals, making them especially suitable for language modeling. We now focus on the AR formulation that underlies most LLMs. Textual data is segmented into a sequence of tokens $\mathcal{X} = \{\mathbf{x}_i, \ldots, \mathbf{x}_1\}$, and a language modeling objective is employed to maximize the likelihood of such data (Radford & Narasimhan, 2018). In other words, we model the joint probability of tokens in the sequence $\mathcal{X}$, through a conditional probability parameterized by $\boldsymbol{\theta}$:

$$p(\mathbf{x}_{i:1}) = p(\mathbf{x}_i \mid \mathbf{x}_{i-1:1}) \ldots p(\mathbf{x}_2 \mid \mathbf{x}_1)\, p(\mathbf{x}_1) = \prod_i \underbrace{p_{\boldsymbol{\theta}}(\mathbf{x}_i \mid \mathbf{x}_{i-1:1})}_{\text{discriminative model}} p_{\boldsymbol{\theta}}(\mathbf{x}_1). \tag{1}$$

What we find interesting about this factorization is that, although it seeks to attain *generative modeling*, i.e., $p(\mathbf{x}_{i:1})$, it actually uses recursively *discriminative classifiers*, parameterized by a transformer network $\boldsymbol{\theta}$, that predicts a discrete distribution of the next token $\mathbf{x}_i$ over the vocabulary $\mathcal{V}$, given previous tokens $\mathbf{x}_{i-1:1}$. This is used to model each conditional probability.

## 4 HOW ENERGY SPILLS IN LLMS

When predicting the token at position $i$, the conditional probability modeled by $\boldsymbol{\theta}$ can be decomposed using the probabilities of the sequences. As a result, the marginal term from step $i$ cancels out with the sequence probability from the decomposition at the previous step $i - 1$, which means we have:

$$p(\mathbf{x}_{i:1}) = \prod_i p_{\boldsymbol{\theta}}(\mathbf{x}_i|\mathbf{x}_{i-1:1}) = \prod_i \frac{p_{\boldsymbol{\theta}}(\mathbf{x}_{i:1})}{p_{\boldsymbol{\theta}}(\mathbf{x}_{i-1:1})} \implies \ldots \frac{p_{\boldsymbol{\theta}}(\mathbf{x}_{i:1})}{\underbrace{p_{\boldsymbol{\theta}}(\mathbf{x}_{i-1:1})}_{\text{step } i}} \overbrace{\frac{p_{\boldsymbol{\theta}}(\mathbf{x}_{i-1:1})}{p_{\boldsymbol{\theta}}(\mathbf{x}_{i-2:1})}}^{\text{step } i-1} \ldots = p(\mathbf{x}_{i:1}).$$

$$\tag{2}$$

This indeed confirms that Eq. (1) results in the correct formulation for language modeling, which is $p(\mathbf{x}_{i:1})$. Following the mathematics, these quantities should cancel out along the sequence, but we will now show that, in practice, *this constraint is not explicitly optimized for, and we can exploit it for hallucination detection*.

## 4.1 INTERPRETING LLMS AS ENERGY-BASED MODELS (EBMS)

Let us continue the expansion from Eq. (2). Writing the conditional as the ratio between the joint distribution in the numerator and the marginal distribution in the denominator, we note that this ratio is actually implemented in LLMs as a softmax classifier that digests the embedding of the prior sentence $\mathbf{x}_{i-1:1}$ and predicts the next token $\mathbf{x}_i$; thus, this chain of equality holds true. We can then apply the "trick" from Grathwohl et al. (2020) as:

$$p_{\boldsymbol{\theta}}(\mathbf{x}_i|\mathbf{x}_{i-1:1}) = \frac{p_{\boldsymbol{\theta}}(\mathbf{x}_{i:1})}{p_{\boldsymbol{\theta}}(\mathbf{x}_{i-1:1})} = \frac{\exp \boldsymbol{\theta}(\mathbf{x}_{i-1:1})\,[\text{id}(\mathbf{x}_i)]}{\sum_{k=1}^{V} \exp \boldsymbol{\theta}(\mathbf{x}_{i-1:1})[k]} \text{ where id}: \{0,1\}^V \mapsto [1, \ldots, V]. \tag{3}$$

id is the map that takes as input a one-hot encoding vector $\mathbf{x}_i$ for a word token at position $i$ in the text and outputs its index in the vocabulary. A typical cross-entropy loss only optimizes with the

supervision provided by the ground-truth token, through the vocabulary index $\text{id}(\mathbf{x}_i)$. This loss ignores all other quantities or constraints related to the complete sequence $\mathcal{X}$, i.e., it ignores all the time steps higher than $i + 1$.

We can write the conditional probability of Eq. (3) as a ratio of two EBMs as:

$$\log p_{\boldsymbol{\theta}}(\mathbf{x}_i|\mathbf{x}_{i-1:1}) = \log \frac{\exp(-E_{\boldsymbol{\theta}}^{\ell}(\mathbf{x}_{i:1}))}{\exp(-E_{\boldsymbol{\theta}}^{m}(\mathbf{x}_{i-1:1}))} \frac{\widetilde{Z}(\boldsymbol{\theta})}{Z(\boldsymbol{\theta})} = -E_{\boldsymbol{\theta}}^{\ell}(\mathbf{x}_{i:1}) + E_{\boldsymbol{\theta}}^{m}(\mathbf{x}_{i-1:1}). \tag{4}$$

Following Zhu et al. (2021), the partition functions simplify since $\log \widetilde{Z}(\boldsymbol{\theta}) = \log Z(\boldsymbol{\theta})$[1].

$E_{\boldsymbol{\theta}}^{\ell}$, $E_{\boldsymbol{\theta}}^{m}$ are computed from the output of the model, but with two big differences: $E_{\boldsymbol{\theta}}^{\ell}$ as a single *logit* extracted using the $\text{id}$ of the sampled token, $E_{\boldsymbol{\theta}}^{m}$ by *marginalizing* over all $\text{id}$s in the vocabulary.

The two energies can be derived from the softmax of the logits, by connecting Eq. (4) and Eq. (3):

$$-\log p_{\boldsymbol{\theta}}(\mathbf{x}_i \mid \mathbf{x}_{i-1:1}) = -\log \left( \frac{\exp(\boldsymbol{\theta}(\mathbf{x}_{i-1:1})[\text{id}(\mathbf{x}_i)])}{\sum_k \exp(\boldsymbol{\theta}(\mathbf{x}_{i-1:1})[k])} \right) = \tag{5}$$

$$= \underbrace{-\boldsymbol{\theta}(\mathbf{x}_{i-1:1})[\text{id}(\mathbf{x}_i)]}_{E_{\boldsymbol{\theta}}^{\ell}(\mathbf{x}_{i:1})} + \underbrace{\log \sum_{k=1}^{V} \exp \boldsymbol{\theta}(\mathbf{x}_{i-1:1})[k]}_{-E_{\boldsymbol{\theta}}^{m}(\mathbf{x}_{i-1:1})} \tag{6}$$

where $\boldsymbol{\theta}(\mathbf{x}_{i-1:1})$ produces the logits over the entire vocabulary $\mathcal{V}$, and $\text{id}(\mathbf{x}_i)$ allows us to extract the logit of the sampled token at decoding step $i$.

We can think of $E_{\boldsymbol{\theta}}^{\ell}(\mathbf{x}_{i:1})$ as the energy of the sampled tokens $\{\mathbf{x}_{i:1}\}$, and $E_{\boldsymbol{\theta}}^{m}(\mathbf{x}_{i-1:1})$ as the energy $E_{\boldsymbol{\theta}}(\mathbf{x}_{i:1})$, marginalized over all possible $\mathbf{x}_i$. Considering the decoding at step $i$ in Eq. (4), we get:

$$E_{\boldsymbol{\theta}}^{\ell}(\mathbf{x}_{i:1}) = -\boldsymbol{\theta}(\mathbf{x}_{i-1:1})[\text{id}(\mathbf{x}_i)], \quad E_{\boldsymbol{\theta}}^{m}(\mathbf{x}_{i-1:1}) = -\log \sum_{k=1}^{V} \exp \boldsymbol{\theta}(\mathbf{x}_{i-1:1})[k]. \tag{7}$$

Using the chain rule and Eq. (6), we can write the negative log-likelihood in terms of energies as:

$$-\log p(\mathbf{x}_{N:1}) = -\log \prod_i p_{\boldsymbol{\theta}}(\mathbf{x}_i|\mathbf{x}_{i-1:1}) = \sum_i E_{\boldsymbol{\theta}}^{\ell}(\mathbf{x}_{i:1}) - E_{\boldsymbol{\theta}}^{m}(\mathbf{x}_{i-1:1})$$

without considering the base case $p_{\boldsymbol{\theta}}(\mathbf{x}_1)$. Now, if we develop the above equation as done for Eq. (2), we write the total energy of a sequence of length $N$ as $E_{\boldsymbol{\theta}}(\mathbf{x}_{N:1})$. Observe that the two energies, not interacting at the same step but at steps $i$ and $i - 1$, **should be equal, but they are measured in the LLM at different generation steps and from different components.**

$$E_{\boldsymbol{\theta}}(\mathbf{x}_{N:1}) = \sum_{i=1}^{N-1} E_{\boldsymbol{\theta}}^{\ell}(\mathbf{x}_{i+1:1}) - E_{\boldsymbol{\theta}}^{m}(\mathbf{x}_{i:1}) = \ldots \overbrace{E_{\boldsymbol{\theta}}^{\ell}(\mathbf{x}_{i+1:1})}^{} \underbrace{-E_{\boldsymbol{\theta}}^{m}(\mathbf{x}_{i:1}) + E_{\boldsymbol{\theta}}^{\ell}(\mathbf{x}_{i:1})}_{\Delta E_{\boldsymbol{\theta}}(\mathbf{x}_{i:1})} \overbrace{-E_{\boldsymbol{\theta}}^{m}(\mathbf{x}_{i-1:1})}^{} \ldots$$

(with labels "timestep $i+1$" and "timestep $i$" above the braces)

At timestep $i + 1$, first $-E_{\boldsymbol{\theta}}^{m}(\mathbf{x}_{i:1})$ is measured, taking the denominator in the softmax as in the right part of Eq. (6), whereas at timestep $i$, the second $E_{\boldsymbol{\theta}}^{\ell}(\mathbf{x}_{i:1})$ is taken, reading the logit in the softmax, left part of Eq. (6). We thus define the discrepancy between the two quantities as **spilled energy**:

> **Definition 4.1** (Spilled Energy $\Delta E_{\boldsymbol{\theta}}(\mathbf{x}_{i:1})$). The spilled energy in an LLM is the difference between two energies that, in principle, should be equal, but given that they are measured i) at different time steps ii) in different components, could be different.
>
> $$\Delta E_{\boldsymbol{\theta}}(\mathbf{x}_{i:1}) \triangleq -E_{\boldsymbol{\theta}}^{m}(\mathbf{x}_{i:1}) + E_{\boldsymbol{\theta}}^{\ell}(\mathbf{x}_{i:1}) = \underbrace{-\log \sum_k \exp(\boldsymbol{\theta}(\mathbf{x}_{i:1})[k])}_{\text{timestep } i+1} + \underbrace{\boldsymbol{\theta}(\mathbf{x}_{i-1:1})[\text{id}(\mathbf{x}_i)]}_{\text{timestep } i}$$
>
> $$\tag{8}$$

Since both terms on the right side should be equal to $E_{\boldsymbol{\theta}}(\mathbf{x}_{i:1})$, delta values should always be zero when we are correctly modeling the energy at timestep $i$. A shorter explanation for why spilled energy needs to be zero is given in Appendix A.3.

---

[1]For a formal proof, please see Appendix A.1.

## 4.2 DETECTING HALLUCINATIONS WITH SPILLED ENERGY

EBMs have previously been used to assess neural network credibility (Liu et al., 2020), and calibration for LLMs has been explored by the Anthropic team (Kadavath et al., 2022b). However, dominant training-free baselines such as logits or "$p(\text{true})$" remain weak. We likewise adopt a training-free approach, but rely on Eq. (8) and its variants as discriminants.

We feed the prompt $\{\mathbf{x}_{i-1}, \ldots, \mathbf{x}_1\}$ to the LLM $\boldsymbol{\theta}$ and obtain the completion $\{\mathbf{x}_N, \ldots, \mathbf{x}_i\}$. Following Orgad et al. (2025), we focus on the "exact answer" tokens—those in $[i+1, N]$ that contain the precise answer (e.g., Rome in Fig. 1), denoted $[u, w] \subseteq [i+1, N]$. For instance, it would be the tokens associated with Rome in the question in Fig. 1. We identify this span by prompting the LLM for a brief answer. When the answer spans multiple tokens, we apply a pooling strategy, which we ablate in Section 5. We propose measuring two values that correlate well with hallucinations:

1. Marginal energy $E_{\boldsymbol{\theta}}^m(\mathbf{x}_{i:1})$;
2. Spilled energy $\Delta E_{\boldsymbol{\theta}}(\mathbf{x}_{i:1})$ by definition of Eq. (8).

We also attempt to combine the two metrics into scaled spilled energy $\Delta E_s$, where the spilled energy is multiplied by the absolute value of the marginal energy as $\Delta E_s(\mathbf{x}_{i:1}) = |E_{\boldsymbol{\theta}}^m(\mathbf{x}_{i:1})| \Delta E_{\boldsymbol{\theta}}(\mathbf{x}_{i:1})$. The metrics proposed here are independent, new for LLMs, and can all be tested efficiently. These measures can be computed over the full sequence, but for error detection, as discussed in Section 5.2, we must extract the values in the localized exact interval $[u, w]$ to avoid false positives. Note that $E_{\boldsymbol{\theta}}^\ell(\mathbf{x}_{i:1})$ is the classic baseline which in literature is referred to as "logits" or "logits confidence".

## 5 EXPERIMENTS

To evaluate spilled energy, we consider two complementary settings. First, a controlled synthetic environment, where we generate both correct and incorrect multi-digit arithmetic solutions. Second, established real-world benchmarks, where errors arise naturally across diverse reasoning and comprehension tasks. Together, these experiments test whether insights from the clean synthetic setup transfer to the complexity of open-domain language understanding.

### 5.1 SPILLED ENERGY UNDER SYNTHETIC ARITHMETIC

**Experimental Setting.** We first evaluate spilled energy in a controlled setting: multi-digit arithmetic problems with more than 14 digits. For each instance, we generate both correct and incorrect solutions. We tested three different LLMs: Llama-3 8B (Dubey et al.), Qwen-3 8B (Qwen-Team), and Mistral-7B-Instruct v0.3 (Jiang et al.). Incorrect solutions are obtained by introducing random numerical errors of varying magnitude. Specifically, we define three error ranges that differ in their difficulty of detection:

◇ **Easy**: random offset in the range $[1000, 10000]$, which are typically easier to identify.
◇ **Medium**: random offset in the range $[100, 1000]$, where detection requires closer inspection.
◇ **Hard**: random offset in $[1, 10]$, much harder to detect since they appear plausible at first glance.

This design allows us to systematically probe whether spilled energy can distinguish between correct and incorrect generations across different levels of error subtlety.

**Results.** We observe that spilled energy values separate correct from incorrect solutions with high reliability across all error ranges and across all LLMs. In particular, spilled energy consistently assigns lower values to correct generations and higher values to incorrect ones, producing a clear margin of separation. Compared to standard baselines such as *logits*, spilled energy achieves superior discriminative power, especially for errors in the more challenging range $[1, 10]$, see Fig. 3. We offer more results in Fig. 5. Larger, better-detailed ROC and histograms are in Figs. 6 and 7 respectively.

### 5.2 CROSS-DATASET RESULTS IN REAL-WORLD BENCHMARKS

**Experimental Setting.** We evaluate our methods on a diverse set of established NLP benchmarks, including Math (Hendrycks et al.), TriviaQA (Joshi et al.), HotpotQA (Yang et al.), Winogrande

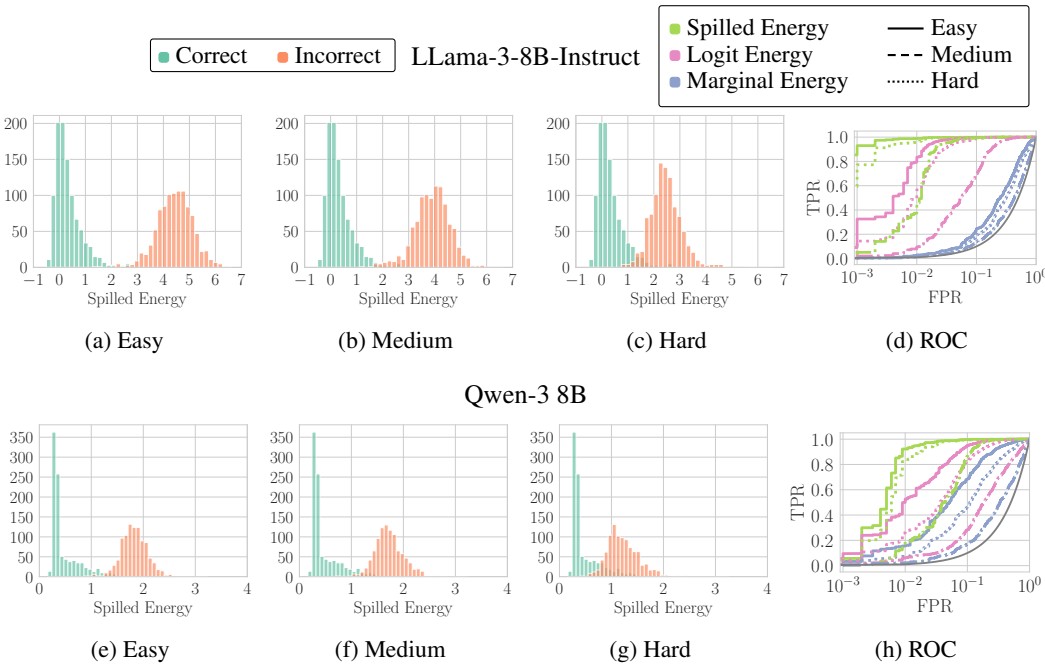

Figure 3: Histograms of Spilled Energy values across models (rows) on Math Sums with different error ranges in the answer (columns, decreasing range left to right, making it harder to detect errors). All sums are performed on 13-digit integers. In the fourth column, we show ROC curves for Hallucination Detection across the error ranges (colors) and methods (line styles).

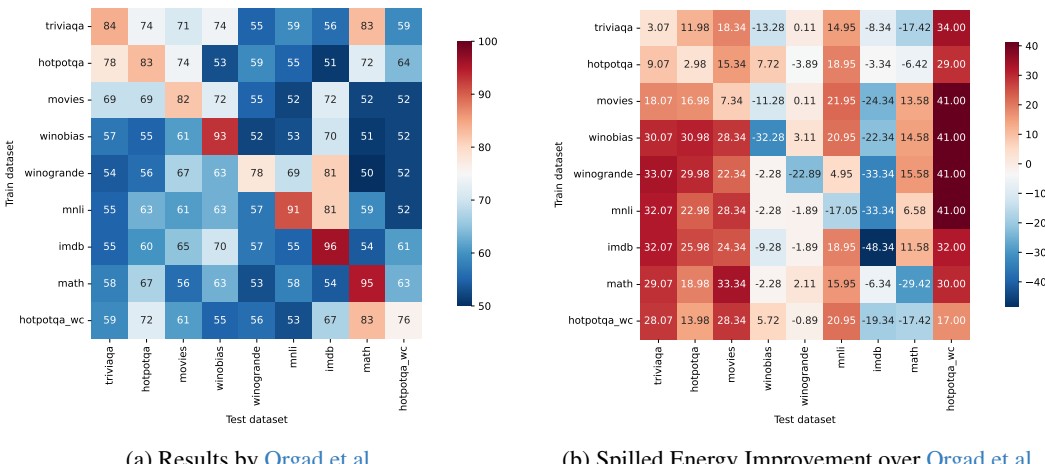

(a) Results by Orgad et al.    (b) Spilled Energy Improvement over Orgad et al.

Figure 4: (a) AuROC performance as percentages of probing classifiers on exact answer tokens by Orgad et al. for LlaMA-3-Instruct. (b) depicts the performance difference between our Spilled $\Delta E$ with Min pooling and theirs. Positive values indicate cases where Spilled $\Delta E$ outperforms Orgad et al.. This comparison highlights the generalization capabilities of our method, compared to probing classifiers. Legend: low performance █████ high performance.

(Sakaguchi et al.), Winobias (Zhao et al.), Movies (Orgad et al.), MNLI (Williams et al.) and IMDB (Maas et al.). These datasets span a wide range of reasoning and error-detection tasks, allowing us to test whether the patterns observed in the synthetic arithmetic setting extend to real-world, open-domain scenarios. Here too, we evaluate multiple LLMs that are either instruction-aligned or not aligned, such as LLaMA-3 (Dubey et al.), and Mistral (Jiang et al.). As emphasized by Orgad et al., it is essential to first localize the tokens most relevant to the final answer before applying error detection. Since exact answer tokens may consist of multiple tokens, we further adopt a pooling

Table 1: Hallucination detection performance, in terms of AuROC, across nine benchmarks and four different LLMs. We measure the generalization across all tasks by computing the average.

| | Pool | HotpotQA | HotpotQA-WC | IMDB | Math | MNLI | Movies | TriviaQA | Winobias | Winogrande | Average |
|---|---|---|---|---|---|---|---|---|---|---|---|
| | | | | | | LLaMA-Instruct Dubey et al. (2024) | | | | | |
| $p$(true) | — | $58.31_{\pm0.32}$ | $51.66_{\pm1.05}$ | $50.72_{\pm1.20}$ | $49.53_{\pm2.16}$ | $52.33_{\pm0.98}$ | $59.30_{\pm0.85}$ | $45.99_{\pm0.51}$ | $45.47_{\pm1.58}$ | $48.33_{\pm0.68}$ | $51.29_{\pm04.86}$ |
| Orgad et al. | Mean | $66.56_{\pm9.10}$ | $59.00_{\pm8.14}$ | $69.78_{\pm14.76}$ | $66.56_{\pm17.04}$ | $60.56_{\pm12.53}$ | $66.44_{\pm8.06}$ | $63.22_{\pm11.11}$ | $67.33_{\pm11.97}$ | $58.00_{\pm7.79}$ | $64.16_{\pm03.90}$ |
| Logit $E^\ell$ | Max | $72.85_{\pm2.12}$ | $91.11_{\pm1.52}$ | $42.08_{\pm5.07}$ | $57.81_{\pm3.82}$ | $25.52_{\pm3.00}$ | $43.97_{\pm1.38}$ | $68.89_{\pm1.96}$ | $39.95_{\pm2.41}$ | $49.40_{\pm2.16}$ | $54.62_{\pm18.97}$ |
| Marginal $E^m$ | Max | $76.72_{\pm1.38}$ | $30.74_{\pm3.45}$ | $\mathbf{85.63}_{\pm2.39}$ | $27.08_{\pm5.06}$ | $\mathbf{89.90}_{\pm1.25}$ | $\mathbf{96.17}_{\pm0.63}$ | $80.13_{\pm1.87}$ | $57.67_{\pm2.94}$ | $47.47_{\pm1.83}$ | $65.72_{\pm24.39}$ |
| Marginal $E^m$ | Min | $75.91_{\pm1.62}$ | $\mathbf{97.57}_{\pm0.75}$ | $14.37_{\pm2.39}$ | $\mathbf{70.55}_{\pm2.43}$ | $61.21_{\pm3.24}$ | $72.21_{\pm1.60}$ | $73.38_{\pm1.86}$ | $47.19_{\pm2.71}$ | $53.98_{\pm2.30}$ | $62.93_{\pm21.89}$ |
| Spilled $\Delta E_s$ | Max | $53.65_{\pm1.40}$ | $36.28_{\pm2.99}$ | $55.80_{\pm4.32}$ | $35.44_{\pm3.41}$ | $58.81_{\pm2.58}$ | $70.30_{\pm1.49}$ | $48.70_{\pm2.44}$ | $36.53_{\pm2.98}$ | $44.32_{\pm1.70}$ | $48.87_{\pm11.26}$ |
| Spilled $\Delta E$ | Min | $\mathbf{85.98}_{\pm1.09}$ | $\mathbf{93.00}_{\pm1.61}$ | $47.66_{\pm4.06}$ | $65.58_{\pm3.02}$ | $73.95_{\pm1.97}$ | $89.34_{\pm1.04}$ | $\mathbf{87.07}_{\pm1.33}$ | $60.72_{\pm2.74}$ | $55.11_{\pm2.05}$ | $\mathbf{73.16}_{\pm15.64}$ |
| | | | | | | LLaMA Dubey et al. (2024) | | | | | |
| $p$(true) | — | $52.83_{\pm0.71}$ | $49.33_{\pm0.86}$ | $52.30_{\pm0.58}$ | $58.63_{\pm1.26}$ | $53.78_{\pm0.70}$ | $60.76_{\pm0.69}$ | $62.94_{\pm0.51}$ | $50.02_{\pm1.24}$ | $53.47_{\pm0.54}$ | $54.90_{\pm04.77}$ |
| Orgad et al. | Mean | $61.22_{\pm9.95}$ | $56.78_{\pm8.70}$ | $\mathbf{72.67}_{\pm13.91}$ | $69.67_{\pm15.07}$ | $60.33_{\pm13.77}$ | $\mathbf{64.00}_{\pm8.40}$ | $66.44_{\pm8.20}$ | $\mathbf{60.89}_{\pm12.60}$ | $\mathbf{53.56}_{\pm4.36}$ | $62.84_{\pm05.71}$ |
| Logit $E^\ell$ | Max | $53.47_{\pm2.13}$ | $49.02_{\pm1.79}$ | $48.27_{\pm1.32}$ | $57.38_{\pm6.09}$ | $91.76_{\pm0.91}$ | $57.42_{\pm1.43}$ | $52.77_{\pm2.58}$ | $50.74_{\pm1.51}$ | $51.17_{\pm1.83}$ | $56.89_{\pm12.70}$ |
| Marginal $E^m$ | Max | $78.00_{\pm1.30}$ | $76.90_{\pm1.09}$ | $48.29_{\pm1.16}$ | $68.77_{\pm8.33}$ | $10.93_{\pm1.42}$ | $\mathbf{80.70}_{\pm1.98}$ | $67.49_{\pm1.69}$ | $51.91_{\pm2.32}$ | $51.28_{\pm2.47}$ | $59.36_{\pm20.69}$ |
| Marginal $E^m$ | Min | $58.39_{\pm2.79}$ | $59.20_{\pm1.95}$ | $51.71_{\pm1.16}$ | $34.13_{\pm8.78}$ | $\mathbf{97.42}_{\pm0.51}$ | $50.37_{\pm2.43}$ | $69.88_{\pm1.40}$ | $49.05_{\pm2.20}$ | $49.00_{\pm2.30}$ | $57.68_{\pm16.75}$ |
| Spilled $\Delta E_s$ | Min | $77.75_{\pm1.52}$ | $79.44_{\pm2.05}$ | $43.39_{\pm1.82}$ | $72.87_{\pm6.10}$ | $\mathbf{99.97}_{\pm0.08}$ | $61.56_{\pm2.95}$ | $77.55_{\pm1.62}$ | $52.34_{\pm2.57}$ | $48.17_{\pm1.62}$ | $68.12_{\pm17.15}$ |
| Spilled $\Delta E$ | Min | $\mathbf{79.04}_{\pm1.78}$ | $\mathbf{80.83}_{\pm1.87}$ | $43.22_{\pm1.67}$ | $\mathbf{74.36}_{\pm5.54}$ | $\mathbf{99.97}_{\pm0.08}$ | $61.97_{\pm2.81}$ | $\mathbf{78.54}_{\pm1.57}$ | $52.11_{\pm2.58}$ | $48.21_{\pm1.62}$ | $\mathbf{68.69}_{\pm17.48}$ |
| | | | | | | Mistral-Instruct Jiang et al. (2023) | | | | | |
| $p$(true) | — | $56.67_{\pm0.80}$ | $53.41_{\pm0.68}$ | $48.84_{\pm0.78}$ | $51.63_{\pm1.29}$ | $54.93_{\pm0.53}$ | $60.64_{\pm0.47}$ | $63.59_{\pm0.57}$ | $56.34_{\pm0.92}$ | $56.92_{\pm0.57}$ | $55.88_{\pm04.45}$ |
| Orgad et al. | Mean | $64.78_{\pm10.56}$ | $56.78_{\pm7.95}$ | $\mathbf{82.67}_{\pm11.63}$ | $\mathbf{68.78}_{\pm11.43}$ | $64.22_{\pm12.12}$ | $64.89_{\pm11.55}$ | $65.44_{\pm12.10}$ | $\mathbf{61.00}_{\pm12.23}$ | $\mathbf{61.44}_{\pm11.31}$ | $65.56_{\pm06.84}$ |
| Logit $E^\ell$ | Max | $77.24_{\pm1.66}$ | $83.84_{\pm1.66}$ | $22.28_{\pm2.54}$ | $57.67_{\pm3.29}$ | $78.98_{\pm1.58}$ | $76.89_{\pm1.49}$ | $80.35_{\pm1.88}$ | $45.53_{\pm2.60}$ | $48.17_{\pm1.97}$ | $63.44_{\pm19.99}$ |
| Marginal $E^m$ | Max | $64.63_{\pm1.97}$ | $33.42_{\pm1.90}$ | $81.33_{\pm2.32}$ | $26.52_{\pm2.28}$ | $17.62_{\pm1.20}$ | $\mathbf{86.60}_{\pm1.20}$ | $65.46_{\pm2.25}$ | $56.41_{\pm4.44}$ | $51.14_{\pm1.71}$ | $53.68_{\pm22.53}$ |
| Marginal $E^m$ | Min | $87.58_{\pm1.35}$ | $\mathbf{97.94}_{\pm0.62}$ | $18.67_{\pm2.27}$ | $67.58_{\pm3.37}$ | $\mathbf{97.96}_{\pm0.55}$ | $84.90_{\pm1.37}$ | $87.75_{\pm1.73}$ | $49.19_{\pm3.97}$ | $48.49_{\pm1.86}$ | $71.12_{\pm25.68}$ |
| Spilled $\Delta E_s$ | Max | $49.13_{\pm2.50}$ | $36.37_{\pm2.40}$ | $46.45_{\pm2.56}$ | $29.05_{\pm2.57}$ | $53.79_{\pm1.55}$ | $55.24_{\pm2.17}$ | $46.73_{\pm1.98}$ | $53.30_{\pm3.66}$ | $51.20_{\pm1.84}$ | $46.81_{\pm08.24}$ |
| Spilled $\Delta E$ | Min | $\mathbf{91.12}_{\pm1.10}$ | $97.47_{\pm0.78}$ | $59.77_{\pm2.57}$ | $66.63_{\pm3.46}$ | $95.95_{\pm0.83}$ | $\mathbf{94.99}_{\pm0.93}$ | $\mathbf{91.75}_{\pm1.01}$ | $50.74_{\pm3.15}$ | $49.00_{\pm1.92}$ | $\mathbf{77.49}_{\pm19.42}$ |
| | | | | | | Mistral Jiang et al. (2023) | | | | | |
| $p$(true) | — | $54.21_{\pm0.76}$ | $51.68_{\pm0.76}$ | $50.40_{\pm0.50}$ | $45.86_{\pm2.05}$ | $51.94_{\pm0.50}$ | $49.12_{\pm0.63}$ | $58.00_{\pm0.67}$ | $53.76_{\pm1.17}$ | $47.29_{\pm0.55}$ | $51.36_{\pm03.73}$ |
| Orgad et al. | Mean | $61.78_{\pm9.27}$ | $57.44_{\pm6.95}$ | $\mathbf{76.22}_{\pm12.82}$ | $65.78_{\pm15.27}$ | $56.67_{\pm11.83}$ | $64.22_{\pm8.91}$ | $64.33_{\pm10.40}$ | $\mathbf{58.00}_{\pm12.29}$ | $\mathbf{54.56}_{\pm4.36}$ | $62.11_{\pm06.21}$ |
| Logit $E^\ell$ | Max | $49.54_{\pm1.42}$ | $52.47_{\pm1.61}$ | $32.72_{\pm2.89}$ | $57.21_{\pm3.89}$ | $92.49_{\pm1.15}$ | $30.52_{\pm2.00}$ | $39.73_{\pm2.03}$ | $46.53_{\pm3.80}$ | $44.41_{\pm2.42}$ | $49.51_{\pm17.28}$ |
| Marginal $E^m$ | Max | $83.57_{\pm1.13}$ | $86.83_{\pm1.70}$ | $45.31_{\pm2.49}$ | $62.26_{\pm1.31}$ | $96.03_{\pm0.83}$ | $\mathbf{99.27}_{\pm0.69}$ | $\mathbf{92.26}_{\pm1.31}$ | $51.31_{\pm3.35}$ | $54.49_{\pm2.48}$ | $74.59_{\pm19.91}$ |
| Marginal $E^m$ | Min | $\mathbf{87.52}_{\pm1.31}$ | $\mathbf{90.91}_{\pm1.58}$ | $54.69_{\pm2.49}$ | $\mathbf{86.21}_{\pm1.96}$ | $\mathbf{98.80}_{\pm0.35}$ | $94.41_{\pm0.62}$ | $83.66_{\pm2.16}$ | $52.15_{\pm1.74}$ | $46.37_{\pm2.02}$ | $\mathbf{77.19}_{\pm19.05}$ |
| Spilled $\Delta E_s$ | Max | $60.54_{\pm1.81}$ | $60.18_{\pm1.84}$ | $43.47_{\pm2.76}$ | $71.93_{\pm3.62}$ | $45.94_{\pm2.40}$ | $78.84_{\pm1.53}$ | $67.92_{\pm1.32}$ | $57.24_{\pm3.72}$ | $51.88_{\pm1.90}$ | $59.77_{\pm11.08}$ |
| Spilled $\Delta E$ | Min | $84.24_{\pm1.18}$ | $83.74_{\pm1.41}$ | $57.43_{\pm2.99}$ | $78.26_{\pm2.93}$ | $96.69_{\pm0.62}$ | $84.47_{\pm1.17}$ | $81.27_{\pm1.83}$ | $50.62_{\pm1.72}$ | $48.72_{\pm1.75}$ | $73.94_{\pm16.18}$ |

strategy across the localized span to obtain a final score per sentence. We compare spilled and marginal energy against baselines such as the probing classifiers of Orgad et al., logit confidence of Varshney et al. and $p$(true) of Kadavath et al..

**Ablation of the exact answer token.** We provide an ablation experiment on the impact of selecting the exact answer tokens. Table 2 reports average AuROC over 9 benchmarks and 4 LLMs with the exact answer, along with another column that offers the improvement provided by using the exact answer. Like prior work, we confirm that searching for the exact answer provides a notable boost: the improvement is very pronounced ($\sim 24\%$) for spilled and marginal energy, while the logit baseline receives a modest increase of $9\%$.

**Cross-dataset results.** We next evaluate in the more general setting of cross-dataset transfer, which better reflects real-world usage. For methods requiring training, we report the average performance on each dataset when trained separately on each of the other datasets (e.g., performance on IMDB is the average accuracy of classifiers trained on each of the other nine datasets). Fig. 4 shows a confusion matrix of cross-dataset performance, where the rows represent the training dataset and the columns represent the testing dataset, and where red indicates good performance and blue indicates low accuracy. The model tested is LlaMA-3-Instruct. Fig. 4a shows that probing classifiers, as soon as they go out-of-distribution from the training dataset, perform only marginally better than random guessing. The sharp drop observed in the off-diagonal elements supports our premise that this standard, in-distribution setup significantly overestimates the utility of trained probes for broad LLM deployment. Meanwhile, Fig. 4b displays the improvement of Spilled $\Delta E$ over the probing classifier, where a positive red result means improvement of our method. Ours exhibits greater performance across most datasets without requiring training. The generalization is proved with a strong increment over the off-diagonal. Moreover, in some cases, such as TriviaQA, HotpotQA, and Movies, we have improvements *even on the diagonal*. Additional confusion matrices are available in Appendix D.3.

Table 1 summarizes results across nine benchmarks. The result reported in each cell is the average of the accuracies of Fig. 4a within a column. Spilled energy consistently outperforms *logit* confidence, and substantially surpasses the probing classifiers of Orgad et al. (2025). While this latter performs well when trained and tested on the same dataset, their performance drops sharply under cross-dataset evaluation, as reflected in their higher standard deviations. By contrast, ours requires no training and

Table 3: Hallucination detection performance on the Gemma Model Instruct for different parameters of the model, 1B and 4B.

| | Pool | IMBD | Movies | TriviaQA | Winogrande | Winobias | MNLI | Math | HotpotQA | HotpotQA-WC | Average |
|---|---|---|---|---|---|---|---|---|---|---|---|
| | | | | | Gemma-Instruct 4B Kamath et al. (2025) | | | | | | |
| Logit $E^\ell$ | Max | $50.09_{\pm0.45}$ | $60.88_{\pm3.96}$ | $53.95_{\pm2.10}$ | $49.77_{\pm0.15}$ | $\mathbf{54.43}_{\pm2.80}$ | $27.00_{\pm2.16}$ | $78.64_{\pm3.47}$ | $62.84_{\pm1.97}$ | $64.49_{\pm2.02}$ | $55.79_{\pm13.24}$ |
| Marginal $E^m$ | Max | $49.14_{\pm2.70}$ | $83.02_{\pm1.56}$ | $84.14_{\pm1.39}$ | $\mathbf{51.49}_{\pm1.97}$ | $47.97_{\pm1.80}$ | $\mathbf{100.00}_{\pm0.00}$ | $74.57_{\pm3.60}$ | $83.70_{\pm0.77}$ | $\mathbf{85.95}_{\pm2.03}$ | $73.33_{\pm17.94}$ |
| Marginal $E^m$ | Min | $50.86_{\pm2.70}$ | $51.29_{\pm3.30}$ | $55.33_{\pm1.80}$ | $48.12_{\pm1.89}$ | $51.91_{\pm2.10}$ | $99.01_{\pm0.50}$ | $76.03_{\pm3.27}$ | $62.59_{\pm1.49}$ | $71.84_{\pm2.72}$ | $63.00_{\pm15.75}$ |
| Spilled $\Delta E_s$ | Max | $\mathbf{50.89}_{\pm1.65}$ | $50.77_{\pm5.72}$ | $56.08_{\pm2.48}$ | $50.59_{\pm1.72}$ | $53.53_{\pm2.81}$ | $95.61_{\pm0.56}$ | $43.94_{\pm3.21}$ | $50.87_{\pm1.87}$ | $51.21_{\pm1.68}$ | $55.94_{\pm14.35}$ |
| Spilled $\Delta E$ | Min | $\mathbf{50.89}_{\pm1.65}$ | $86.13_{\pm4.28}$ | $89.01_{\pm1.06}$ | $50.18_{\pm1.97}$ | $53.10_{\pm3.05}$ | $99.66_{\pm0.21}$ | $\mathbf{82.29}_{\pm2.46}$ | $\mathbf{89.10}_{\pm1.75}$ | $82.70_{\pm1.35}$ | $\mathbf{75.89}_{\pm17.98}$ |
| | | | | | Gemma-Instruct 1B Kamath et al. (2025) | | | | | | |
| Logit $E^\ell$ | Max | $46.33_{\pm0.82}$ | $48.12_{\pm11.45}$ | $58.89_{\pm1.61}$ | $50.50_{\pm2.45}$ | $53.49_{\pm3.71}$ | $49.28_{\pm2.12}$ | $65.12_{\pm6.62}$ | $62.24_{\pm3.62}$ | $75.67_{\pm1.96}$ | $56.63_{\pm9.13}$ |
| Marginal $E^m$ | Max | $45.42_{\pm1.78}$ | $94.15_{\pm8.44}$ | $83.66_{\pm1.82}$ | $50.23_{\pm3.83}$ | $49.93_{\pm1.56}$ | $98.17_{\pm0.39}$ | $64.21_{\pm6.67}$ | $86.87_{\pm1.39}$ | $82.33_{\pm1.27}$ | $72.77_{\pm19.33}$ |
| Marginal $E^m$ | Min | $\mathbf{54.58}_{\pm1.78}$ | $28.93_{\pm14.50}$ | $39.80_{\pm2.54}$ | $49.84_{\pm4.38}$ | $50.39_{\pm1.80}$ | $56.33_{\pm1.60}$ | $63.20_{\pm4.27}$ | $41.58_{\pm2.85}$ | $61.56_{\pm1.61}$ | $49.58_{\pm10.47}$ |
| Spilled $\Delta E_s$ | Max | $45.17_{\pm2.37}$ | $33.27_{\pm11.49}$ | $49.01_{\pm1.67}$ | $52.27_{\pm3.56}$ | $49.91_{\pm2.59}$ | $77.48_{\pm1.92}$ | $40.49_{\pm4.17}$ | $49.18_{\pm3.93}$ | $35.77_{\pm2.13}$ | $48.06_{\pm12.13}$ |
| Spilled $\Delta E$ | Min | $45.02_{\pm2.45}$ | $82.82_{\pm12.91}$ | $80.73_{\pm2.16}$ | $\mathbf{52.48}_{\pm3.75}$ | $49.77_{\pm2.82}$ | $92.93_{\pm1.79}$ | $56.82_{\pm6.90}$ | $85.64_{\pm2.23}$ | $71.86_{\pm1.77}$ | $68.67_{\pm16.84}$ |

generalizes robustly across diverse benchmarks. We observe that instruction-tuned models tend to amplify the margin by which spilled energy outperforms other methods, whereas on non-aligned Mistral, spilled energy may rank slightly behind marginal energy. We also compare pooling strategies and find that min pooling yields the best overall performance across methods. Table 3 shows our method generalizes to Gemma over different LLM size, 1B and 4B.

**Impact of Instruction Tuning.** We observe a difference in the behavior in the base models and their instruction-tuned ones. While instruction-tuning generally improves generation quality, it can degrade the calibration of classical confidence metrics, as described in Huang et al. (2023a); Ho et al. (2025). For instance, examining the average performance in Table 1, the logit baseline $E^\ell_\theta$ decreases from 56.89% to 54.62% for LLaMA-3, indicating that fine-tuning may lead to overconfidence. In contrast, Spilled Energy ($\Delta E_\theta$) consistently benefits from instruction tuning, showing improved detection rates across both LLaMA-3 (68.69% to 73.16%) and Mistral (73.94% to 77.49%).

| | Pool | Average % w/ exact answer | Exact answer increase |
|---|---|---|---|
| Logit $E^\ell$ Orgad et al. | Max | 56.12 | +9.23 |
| | Mean | 63.67 | – |
| Marginal $E^m$ | Min | 67.23 | +20,02 |
| Marginal $E^m$ | Max | 63.34 | +3,62 |
| Spilled $\Delta E$ | Min | **73.32** | **+24.06** |

Table 2: Improvements in AuROC with the exact answer. Average across 4 LLMs and 9 benchmarks.

**Variance and Generalization.** A notable observation in Table 1 is the higher standard deviation associated with marginal and spilled energy compared to the probing classifiers in the average column. This variance is not a weakness but a reflection of the method's training-free nature. Since $\Delta E_\theta$ relies on the intrinsic energy landscape of the LLM, its magnitude and sensitivity are naturally dependent on the specific domain (e.g., the sharp energy peaks in *Math* and *HotpotQA* versus the flatter distributions in *Winobias* and *IMDB*). Probing classifiers, by contrast, have high-variance when cross-testing yet the average of cross-testing results is mostly constant just above random chance ($\approx 62 - 64\%$).

**Limitations.** A current limitation of spilled energy is that it sometimes produces false positives on tokens that are not semantically informative, as shown in Appendix D.2. We observe this effect most prominently on punctuation tokens (e.g., commas, periods) and on words at the beginning of sentences. In these cases, the probability mass over the next token is naturally spread across many plausible options, leading to inflated spilled energy values even in otherwise correct generations. This highlights the importance of accurately identifying the *exact answer tokens*, as detection is most reliable when restricted to the parts of the output that carry the semantic content of the answer.

## 6 CONCLUSION

We reinterpreted the softmax layer of LLMs as an EBM, which lets us define *spilled energy*: the discrepancy between energy values that should be equal across consecutive time steps. We show theoretically and empirically that this discrepancy provides a strong, training-free signal for detecting hallucinations and errors in LLM outputs. Through synthetic arithmetic experiments, we demonstrate that spilled energy reliably separates correct from incorrect generations, outperforming baselines such as logits and marginal energy. Across diverse real-world NLP benchmarks, spilled energy generalizes robustly without requiring additional classifiers or task-specific training, unlike probing methods that struggle with transfer. Overall, spilled energy offers a principled and practical framework for error detection in LLMs and a new perspective on the internal energy dynamics of autoregressive models.

## ETHICS STATEMENT

This work adheres to the ICLR Code of Ethics. Our study focuses on methodological contributions to error and hallucination detection in Large Language Models. We do not train new models or collect additional data; instead, we rely exclusively on publicly available datasets and widely used benchmark models for evaluation.

We note that part of our evaluation includes the Math dataset, which was publicly accessible at the time of experimentation but has since been taken down following a copyright claim. We emphasize that this dataset was used solely for evaluation purposes of our method, and only prior to the date of the takedown. No redistribution of the dataset was made, and our reported results are limited to demonstrating methodological effectiveness.

Our work does not involve personally identifiable information, sensitive content, or human subjects, and does not raise foreseeable risks of harm. We believe the proposed approach contributes positively to research on trustworthy AI by providing a training-free and generalizable framework for error detection in language models.

## REPRODUCIBILITY STATEMENT

We are committed to ensuring the reproducibility of our results. All experimental details, including model configurations, evaluation protocols, and datasets used, are described in the main text and Appendix B. Upon acceptance of this work, we will publicly release the code implementing our method, along with instructions to reproduce all reported experiments. This will allow the community to verify our findings and build upon our work.

## ACKNOWLEDGMENT

This work was supported by projects PNRR MUR PE0000013-FAIR under the MUR National Recovery and Resilience Plan funded by the European Union - NextGenerationEU, PRIN 2022 project 20227YET9B "AdVVent" CUP code B53D23012830006. It was also partially supported by Sapienza research projects D2QNeT and BEAT (Better dEep leArning securiTy) — bando per la ricerca di Ateneo 2024, and via the Seed of ERC grant "MINT.AI" (cup B83C25001040001). This work is additionally supported by the MUR FIS2 grant n. FIS-2023-00942 "NEXUS" (cup B53C25001030001). The work of Hazem Dewidar was carried out while he was enrolled in the Italian National Doctorate on Artificial Intelligence run by Sapienza University of Rome. Computing was supported by CINECA through the Italian SuperComputing Resource Allocation (ISCRA) projects Ge-Di HP10CRPUVC and SLEY HP10CX9CMC.

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

# A  APPENDIX

## A.1  PARTITION FUNCTIONS PROOF USED IN EQ. (4)

We extend the proof of Zhu et al. to the sequence-to-sequence setting by treating next-token prediction as a multi-class classification problem. At step $i$, the input is the prefix $\{\mathbf{x}_{i-1:1}\}$, and the model outputs logits over the vocabulary $\mathcal{V}$ of size $V$. For notational consistency, we define the following energy terms:

$$
\begin{cases}
E_{\boldsymbol{\theta}}^{\ell}(\mathbf{x}_{i:1}) = -\log\big(\exp\big(\boldsymbol{\theta}(\mathbf{x}_{i-1:1})[\mathtt{id}(\mathbf{x}_i)]\big)\big), \\
E_{\boldsymbol{\theta}}^{m}(\mathbf{x}_{i-1:1}) = -\log\Big(\sum_{k=1}^{V} \exp\big(\boldsymbol{\theta}(\mathbf{x}_{i-1:1})[k]\big)\Big).
\end{cases}
\tag{9}
$$

The probability of the sequence up to position $i$ can be expressed as

$$
p_{\boldsymbol{\theta}}(\mathbf{x}_{i:1}) = \frac{\exp(-E_{\boldsymbol{\theta}}^{\ell}(\mathbf{x}_{i:1}))}{Z_{\boldsymbol{\theta}}},
\tag{10}
$$

where $Z_{\boldsymbol{\theta}}$ is the global partition function (normalizing constant), defined over all possible continuations of all prefixes:

$$
Z_{\boldsymbol{\theta}} = \sum_{\mathbf{x}_{i-1:1}} \sum_{\mathbf{x}_i} \exp(\boldsymbol{\theta}(\mathbf{x}_{i-1:1})[\mathtt{id}(\mathbf{x}_i)]) = \sum_{\mathbf{x}_{i-1:1}} \sum_{k=1}^{V} \exp(\boldsymbol{\theta}(\mathbf{x}_{i-1:1})[k]).
\tag{11}
$$

Similarly, the probability of the prefix $\mathbf{x}_{i-1:1}$ can be written using the marginal energy:

$$
p_{\boldsymbol{\theta}}(\mathbf{x}_{i-1:1}) = \frac{\exp(-E_{\boldsymbol{\theta}}^{m}(\mathbf{x}_{i-1:1}))}{\widetilde{Z}_{\boldsymbol{\theta}}},
\tag{12}
$$

where $\widetilde{Z}_{\boldsymbol{\theta}}$ is the corresponding normalizing constant:

$$
\widetilde{Z}_{\boldsymbol{\theta}} = \sum_{\mathbf{x}_{i-1:1}} \exp(-E_{\boldsymbol{\theta}}^{m}(\mathbf{x}_{i-1:1})) = \sum_{\mathbf{x}_{i-1:1}} \exp\left(\log \sum_{k=1}^{V} \exp\big(\boldsymbol{\theta}(\mathbf{x}_{i-1:1})[k]\big)\right).
\tag{13}
$$

By expanding the logarithm in Eq. (13), we obtain

$$
\widetilde{Z}_{\boldsymbol{\theta}} = \sum_{\mathbf{x}_{i-1:1}} \sum_{k=1}^{V} \exp(\boldsymbol{\theta}(\mathbf{x}_{i-1:1})[k]),
\tag{14}
$$

which is identical to Eq. (11). Hence, the two partition functions coincide:

$$
Z_{\boldsymbol{\theta}} = \widetilde{Z}_{\boldsymbol{\theta}}.
\tag{15}
$$

## A.2   THE ROLE OF TEMPERATURE IN SPILLED ENERGY

We now analyze how the temperature parameter $\tau$ affects the definition of spilled energy. Starting from Eq. (3), the probability of the next token under temperature scaling is

$$\log p_\theta(\mathbf{x}_i \mid \mathbf{x}_{i-1:1}) = \log \frac{\exp\left(\frac{1}{\tau}\boldsymbol{\theta}(\mathbf{x}_{i-1:1})[\mathrm{Id}(\mathbf{x}_i)]\right)}{\sum_k \exp\left(\frac{1}{\tau}\boldsymbol{\theta}(\mathbf{x}_{i-1:1})[k]\right)} \tag{16}$$

$$= \frac{1}{\tau}\boldsymbol{\theta}(\mathbf{x}_{i-1:1})[\mathrm{Id}(\mathbf{x}_i)] - \log \sum_k \exp\left(\frac{1}{\tau}\boldsymbol{\theta}(\mathbf{x}_{i-1:1})[k]\right). \tag{17}$$

Accordingly, the spilled energy becomes

$$\Delta E_\theta(\mathbf{x}_{i:1}) = \frac{1}{\tau}\boldsymbol{\theta}(\mathbf{x}_{i-1:1})[\mathrm{Id}(\mathbf{x}_i)] - \log \sum_{k=1}^{|V|} \exp\left(\frac{1}{\tau}\boldsymbol{\theta}(\mathbf{x}_i, \ldots, \mathbf{x}_1)[k]\right). \tag{18}$$

**Limit case $\tau \to \infty$.**   When the temperature tends to infinity, the logits are scaled down towards zero, making all tokens equally likely:

$$\lim_{\tau \to +\infty} \Delta E_\theta(\mathbf{x}_{i:1}) = \lim_{\tau \to \infty} \frac{1}{\tau}\boldsymbol{\theta}(\mathbf{x}_{i-1:1})[\mathrm{Id}(\mathbf{x}_i)] - \log \sum_{k=1}^{|V|} \exp\left(\frac{1}{\tau}\boldsymbol{\theta}(\mathbf{x}_{i-1:1})[k]\right) \tag{19}$$

$$= 0 - \log \sum_{k=1}^{|V|} \exp(0) \tag{20}$$

$$= -\log|V|. \tag{21}$$

Thus, for $\tau \to \infty$ the model degenerates into a uniform random classifier over the vocabulary.

**Interpretation.**   Varying $\tau$ perturbs the balance between the two energy terms, introducing a systematic error in $\Delta E_\theta$. From the perspective of the Boltzmann distribution, scaling by $\frac{1}{\tau}$ corresponds to injecting or removing energy from the system. At high temperatures ($\tau \to \infty$), the system approaches maximum entropy, where all tokens have equal probability. At low temperatures ($\tau \to 0^+$), the distribution collapses onto the maximum logit token, making the model highly deterministic.

**Error accumulation.**   As we generate tokens sequentially, we accumulate deviations in $\Delta E_\theta$:

$$\log p_\theta(\mathbf{x}_{i-1:1}) = \frac{1}{\tau}\boldsymbol{\theta}(\mathbf{x}_{i-1:1})[\mathrm{Id}(\mathbf{x}_i)] - \log \sum_k \exp\left(\frac{1}{\tau}\boldsymbol{\theta}(\mathbf{x}_{i-1:1})[k]\right) + \sum_{j=1}^{i} \Delta E_\theta(\mathbf{x}_{j:1}). \tag{22}$$

Hence, temperature scaling not only modifies the probabilities but also reshapes the cumulative error landscape traced by spilled energy.

## A.3   WHY SPILLED ENERGY SHOULD BE ZERO?

**TL;DR** Consider Eq. (2) in our paper and the simplification that occurs between the two probabilities between step $i$ and step $i-1$: that simplification occurs because the probability in the denominator at step $i$ is the same as the probability in the numerator at step $i-1$ in order to perform language modeling correctly. We measure those inside and LLMs in terms of energy, and the spilled energy is the amount by which they differ.

Please see the definition below. Let us assume a sequence of three tokens $\mathbf{x}_2, \mathbf{x}_1, \mathbf{x}_0$. If we do language modeling with autoregression, minimizing the negative log-likelihood, we have:

$$-\log p(\mathbf{x}_2, \mathbf{x}_1, \mathbf{x}_0) = -\log \underbrace{p(\mathbf{x}_2|\mathbf{x}_1, \mathbf{x}_0)}_{\text{step 2}} p(\mathbf{x}_1|\mathbf{x}_0) p(\mathbf{x}_0)$$

Now, every conditional probability on the right side is implemented with a transformer ending in a softmax discriminative classifier. Eq. (3) and Eq. (6) allow us to re-interpret:

$$\textbf{step 2:} \quad -\log p(\mathbf{x}_2|\mathbf{x}_1, \mathbf{x}_0) = -\log \frac{p(\mathbf{x}_2, \mathbf{x}_1, \mathbf{x}_0)}{p(\mathbf{x}_1, \mathbf{x}_0)} = -\log \left[ \frac{\exp\left(\theta(\mathbf{x}_1, \mathbf{x}_0)[id(\mathbf{x}_2)]\right)}{\sum_k^V \exp\left(\theta(\mathbf{x}_1, \mathbf{x}_0)[k]\right)} \right] = \quad (23)$$

$$= E^\ell(\mathbf{x}_2, \mathbf{x}_1, \mathbf{x}_0) - E^m(\mathbf{x}_1, \mathbf{x}_0). \quad (24)$$

In other words, we reinterpret:

 ⋄ the numerator $p(\mathbf{x}_2, \mathbf{x}_1, \mathbf{x}_0)$ as the energy $E^\ell(\mathbf{x}_2, \mathbf{x}_1, \mathbf{x}_0)$, which is the **logit ($\ell$)** of the softmax at timestep 2;
 ⋄ The denominator as the energy $E^m(\mathbf{x}_1, \mathbf{x}_0)$ obtained with the **marginalization (m)** across the vocabulary $V$. This value can be read "read" simply by taking the denominator of the softmax at timestep 2. Please remember this term.

It is better to indicate them as energies (since they are not probabilities), and given their logarithmic properties, we obtain a difference. We use the notation $l$ for logits and $m$ for marginalization.

Now, **when we go across steps and we connect two-time steps:**

$$\textbf{step 1:} \quad -\log p(\mathbf{x}_1|\mathbf{x}_0) = -\log \frac{p(\mathbf{x}_1, \mathbf{x}_0)}{p(\mathbf{x}_0)} = E^\ell(\mathbf{x}_1, \mathbf{x}_0) - E^m(\mathbf{x}_0).$$

We see that at timestep 1, the value $E^\ell(\mathbf{x}_1, \mathbf{x}_0)$ **appears again, but measured at the logit level.**

In other words, across the time-steps 2 and 1, the quantity $E(\mathbf{x}_1, \mathbf{x}_0)$ is measured twice:

 ⋄ at timestep 2, as the marginalization;
 ⋄ at timestep 1, as the logit.

In the architecture or in the loss, there is no mechanism that forces these quantities to be the same, but they should be equal, given the language modeling objective. This is the same as saying that in Eq. (2), the probabilities across time steps need to cancel out as shown.

In other words, the following:

$$p(\mathbf{x}_2, \mathbf{x}_1, \mathbf{x}_0) = p(\mathbf{x}_2|\mathbf{x}_1, \mathbf{x}_0)p(\mathbf{x}_1|\mathbf{x}_0)p(\mathbf{x}_0)$$

Implies:

$$E(\mathbf{x}_2, \mathbf{x}_1, \mathbf{x}_0) = E^\ell(\mathbf{x}_2, \mathbf{x}_1, \mathbf{x}_0) \underbrace{-E^m(\mathbf{x}_1, \mathbf{x}_0) + E^\ell(\mathbf{x}_1, \mathbf{x}_0)}_{\text{should be zero}} \underbrace{-E^m(\mathbf{x}_0) + E^\ell(\mathbf{x}_0)}_{\text{should be zero}}$$

To model the energy of a sequence $E^\ell(\mathbf{x}_2, \mathbf{x}_1, \mathbf{x}_0)$ correctly, then:

 ⋄ $-E^m(\mathbf{x}_1, \mathbf{x}_0) + E^\ell(\mathbf{x}_1, \mathbf{x}_0) = 0$ (spilled energy at timestep 2 if non-zero)
 ⋄ $-E^m(\mathbf{x}_0) + E^\ell(\mathbf{x}_0) = 0$ (spilled energy at timestep 1 if non-zero)

so that $E(\mathbf{x}_2, \mathbf{x}_1, \mathbf{x}_0) = E^\ell(\mathbf{x}_2, \mathbf{x}_1, \mathbf{x}_0)$.

# B    REPRODUCIBILITY

For comparisons on real-world tasks, we adopt the same experimental setting as Orgad et al. (2025), whose implementation is publicly available at `https://github.com/technion-cs-nlp/LLMsKnow`. This ensures that our baselines and evaluation procedures follow an established and validated protocol.

In addition, we release our codebase, which includes:

⋄ computation of the proposed energy-based measures;

⋄ scripts for reproducing the synthetic arithmetic preliminary experiments;

⋄ example of how our method can be integrated into a benchmarking or production pipeline.

## B.1    EXACT ANSWER TOKEN DETECTION DETAILS

To analyze the **spilled energy** specifically on the tokens carrying the semantic weight of the answer, we must first localize the "exact answer" span $[u, w]$ within the longer generated sequence $\hat{y}$. We adopt the methodology proposed by Orgad et al. (2025), utilizing a combination of heuristics and an auxiliary instruction-tuned LLM to perform this extraction.

**Extraction Strategy**    Depending on the nature of the task, we employ two strategies to identify the exact answer substring $s$:

⋄ **Heuristic Matching:** For tasks with a closed set of possible labels (e.g., classification tasks or multiple-choice QA), we perform string matching to locate the label within the generation.

⋄ **LLM-based Extraction:** For open-ended generation tasks (e.g., TriviaQA, Math), where the answer form varies, we employ an instruction-tuned model (Mistral-7B-Instruct) to extract the short answer from the long-form generation.

**Prompting for Extraction**    Following Orgad et al. (2025), we prompt the auxiliary model with the original question $q$ and the generated long answer $\hat{y}$ using the following template:

```
Prompt for Exact Answer Extraction

Extract from the following long answer the short answer, only the
relevant tokens.  If the long answer does not answer the question,
output NO ANSWER.

Q: [Question 1]
A: [LLM long answer 1]
Exact answer:  [Short exact answer 1]

Q: [Question 2]
A: [LLM long answer that does not answer the question]
Exact answer:  NO ANSWER

Q: [Question]
A: [LLM long answer]
Exact answer:
```

**Verification and Token Mapping**    To ensure robustness, we verify that the extracted string $s$ is a valid substring of the original generation $\hat{y}$. If the extraction is invalid or the model outputs "NO ANSWER," we retry the extraction up to five times. If a valid substring is still not found, the sample is excluded from the analysis to avoid identifying incorrect tokens.

Once the substring $s$ is validated, we map it to the corresponding token indices $[u, w]$ in the original sequence. The spilled energy analysis is then performed specifically over this interval, or pooled across it (e.g., via min-pooling) as described in Section 5.2.

Table 4: Answer Extraction Success Rate across tasks for Mistral-Instruct.

| Dataset | Success Rate (%) |
|---|---|
| TriviaQA | 90.29 |
| HotpotQA | 87.37 |
| Movies | 93.61 |
| MNLI | 92.99 |
| Math | 87.59 |
| HotpotQA-WC | 92.38 |

**Answer Extraction Performance**    For answer localization, we achieve accuracy comparable to the results of Orgad et al. (2025). We report in Table 4 the extraction success rate across the full datasets using Mistral-7B-Instruct. Note that some datasets have been excluded (e.g., IMDB, Winobias, Winogrande) since they have a finite set of possible answers that can be used to easily locate the exact answer within the model's generation.

## C   LLM USAGE

Large language models were used exclusively for text polishing and minor exposition refinements. All substantive research content, methodology, and scientific conclusions were developed entirely by the authors.

# D SUPPLEMENTARY MATERIAL

This supplementary material is intended to complement the main paper by providing further motivation for our assumptions and design choices, as well as additional ablation studies or plots, such as ROCs and histograms, that could not fit in the main paper.

## D.1 ADDITIONAL RESULTS FOR SYNTHETIC ARITHMETIC

In Fig. 5 we augmented Fig. 3 in the main paper, also adding the results for **Mistral-7B-Instruct v0.3** and **LLaMa-3-8B**. The same findings of the figure in the paper also translate to this LLM, meaning that our method generalizes across LLMs.

Fig. 6 and Fig. 7 also extend and provide more details of Fig. 3 in the main paper by showing, respectively, the histograms and the ROC at a better resolution and displayed in different frames. Also, we have added results for Mistral-7B-Instruct v0.3 and LLaMa-3-8B.

## D.2 ADDITIONAL QUALITATIVE RESULTS

In this section, we offer additional results of the detection performance following what is shown in Fig. 1. We report both success cases and failure cases. While it is difficult to draw conclusions and predict when, why, and on which topics spilled energy may work or not, we noticed that it appears to perform reliably on knowledge-based factual content but, at times, exhibits difficulties with reasoning tasks and numerical information, despite working well on math questions, as demonstrated in Section 5.1. Further investigation is required to better understand and validate these patterns.

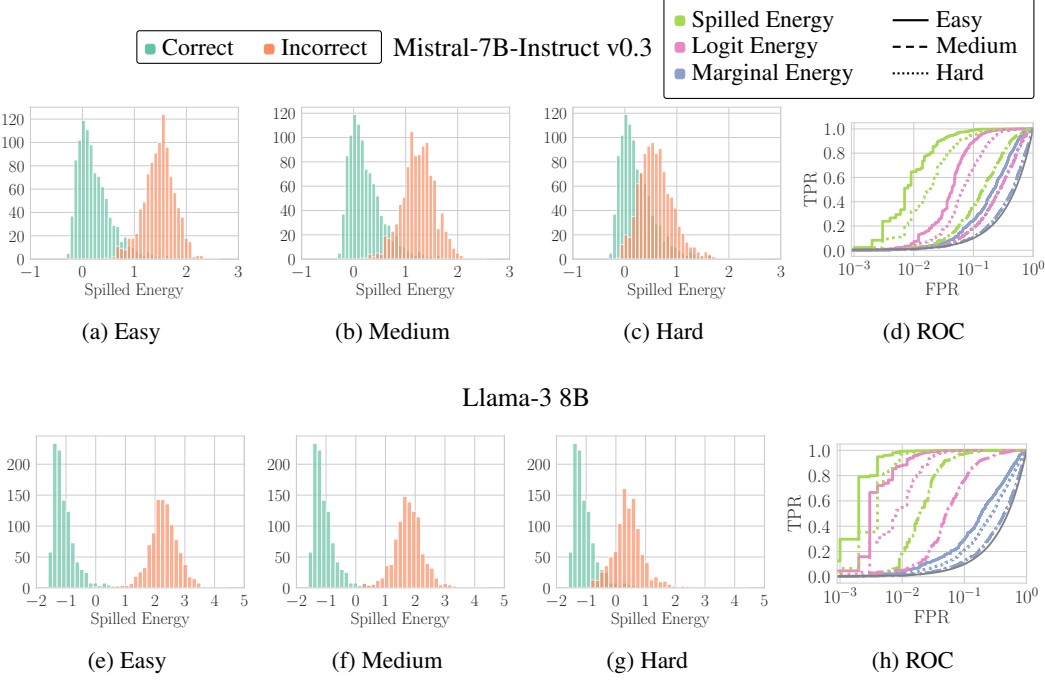

Figure 5: Histograms of Spilled Energy values across models (rows) on Math Sums with different error ranges in the answer (columns, decreasing range left to right, making it harder to detect errors), as described in Section 5.1. In the fourth column, we show ROC curves for Hallucination Detection across the error ranges (colors) and methods (line styles).

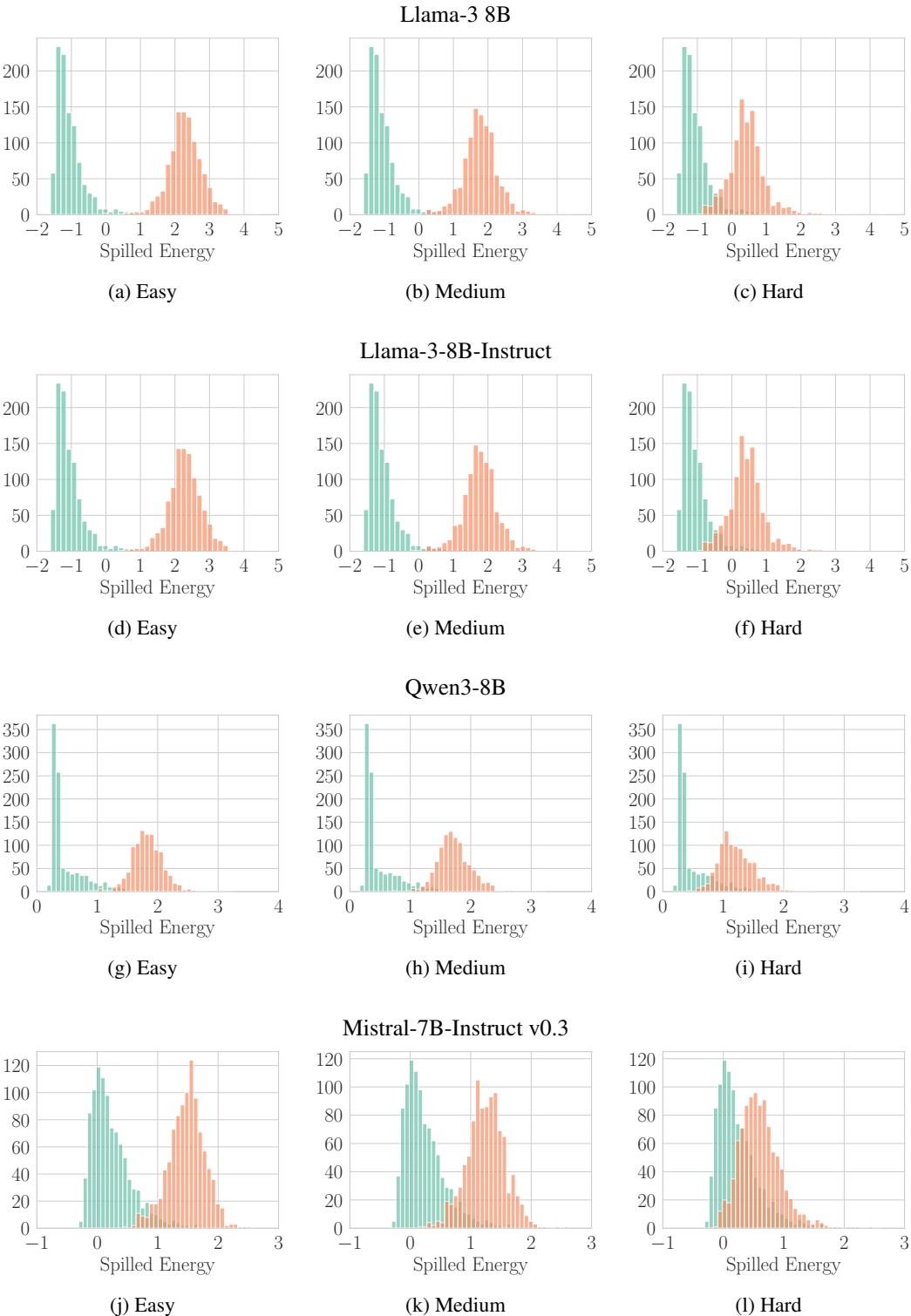

Figure 6: Histograms of Spilled Energy values for ▬ Correct and ▬ Incorrect answers across models on Math Sums, increasing difficulty from left to right. We compute sums on 13-digit integers, for incorrect answers we add a random offset sampled uniformly from the error interval: Easy $\sim \mathcal{U}(1e3, 1e4)$ - Medium $\sim \mathcal{U}(1e2, 1e3)$ - Hard $\sim \mathcal{U}(1, 10)$; for more details see Section 5.1.

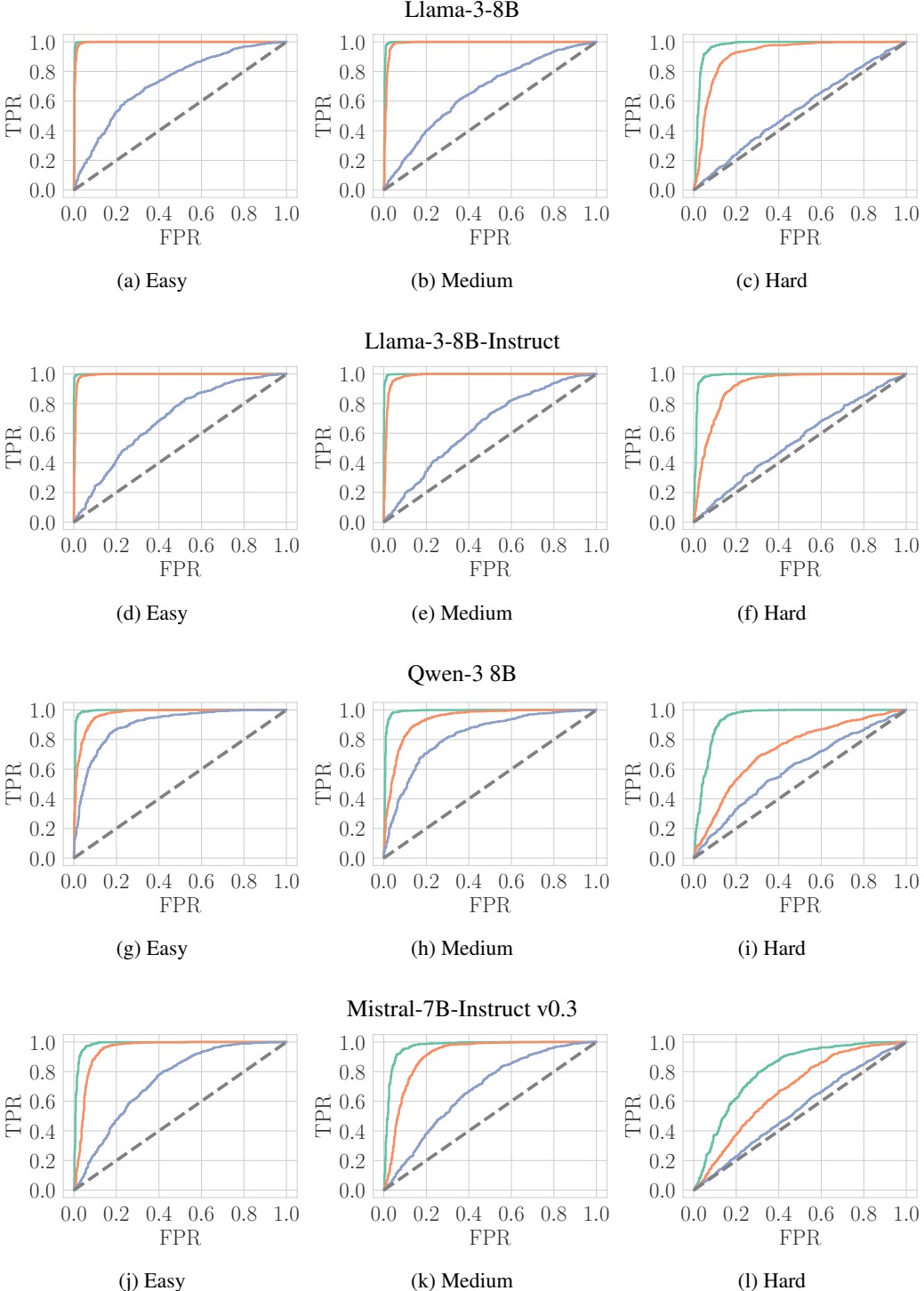

Figure 7: ROC curves for Hallucination Detection across models (rows) on Math Sums with different error ranges in the answer (columns, decreasing range left to right). All sums are performed on 13-digit integers. Legend: **Spilled (ours)** Spilled $\Delta E$   Logit $E^{\ell}$   Marginal $E^{m}$

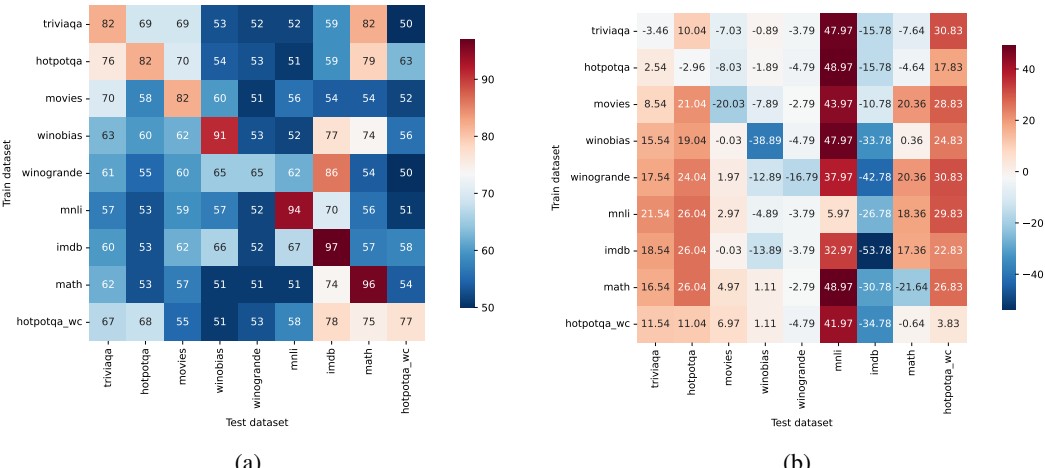

Figure 8: Fig. 8a presents the cross-dataset performance of the method proposed by Orgad et al. (2025) using Llama-3. Fig. 8b depicts the performance difference between their method and our Spilled $\Delta E$ with Min pooling. Positive values indicate cases where Spilled $\Delta E$ outperforms the method of Orgad et al. (2025). All numbers are computed as percentages.

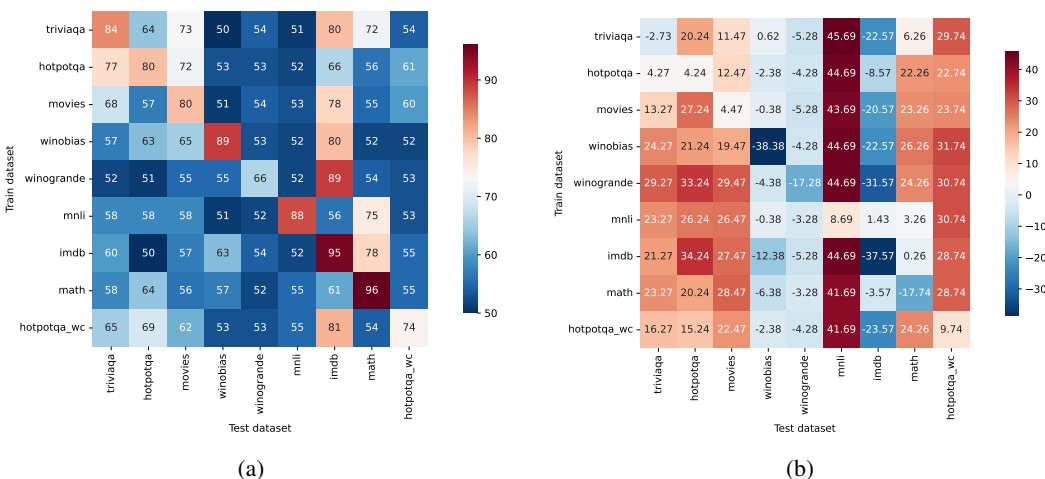

Figure 9: Fig. 9a presents the cross-dataset performance of the method proposed by Orgad et al. (2025) using Mistral. Fig. 9b depicts the performance difference between their method and our Spilled $\Delta E$ with Min pooling. Positive values indicate cases where Spilled $\Delta E$ outperforms the method of Orgad et al. (2025). All numbers are computed as percentages.

|  | Pool | HotpotQA | HotpotQA-WC | IMDB | Math | MNLI | Movies | TriviaQA | Winobias | Winogrande | Average |
|---|---|---|---|---|---|---|---|---|---|---|---|
| | | | | | LLaMA-Instruct | | | | | | |
| Orgad et al. (2025) | Mean | 66.56±9.10 | 59.00±8.14 | *69.78*±14.76 | *66.56*±17.04 | 60.56±12.53 | 66.44±8.06 | 63.22±11.11 | **67.33**±11.97 | **58.00**±7.79 | 64.16±3.90 |
| Spilled ΔE | Min | **85.98**±1.09 | *93.00*±1.61 | 47.66±4.06 | 65.58±3.02 | 73.95±1.97 | 89.34±1.04 | **87.07**±1.33 | 60.72±2.74 | 55.11±2.05 | **73.16**±15.64 |
| Marginal E^m | Max | 76.72±1.38 | 30.74±3.45 | **85.63**±2.39 | 27.08±5.06 | **89.90**±1.25 | **96.17**±0.63 | *80.13*±1.87 | 57.67±2.94 | 47.47±1.83 | *65.72*±24.39 |
| Marginal E^m | Min | 75.91±1.62 | **97.57**±0.75 | 14.37±2.39 | **70.55**±2.43 | 61.21±3.24 | 72.21±1.60 | 73.38±1.86 | 47.19±2.71 | 53.98±2.30 | 62.93±21.89 |
| Logit E^ℓ | Max | 72.85±2.12 | 91.11±1.52 | 42.08±5.07 | 57.81±3.82 | 25.52±3.00 | 43.97±1.38 | 68.89±1.96 | 39.95±2.41 | 49.40±2.16 | 54.62±18.97 |
| Spilled ΔE | Max | 54.34±1.58 | 47.68±2.81 | 52.34±4.06 | 40.33±3.05 | 56.44±2.81 | 68.56±1.87 | 47.54±2.40 | 38.40±2.61 | 44.97±1.51 | 50.07±8.66 |
| | | | | | LLaMA | | | | | | |
| Orgad et al. (2025) | Mean | 61.22±9.95 | 56.78±8.70 | **72.67**±13.91 | 69.67±15.07 | 60.33±13.77 | 64.00±8.40 | 66.44±8.20 | **60.89**±12.60 | **53.56**±4.36 | 62.84±5.71 |
| Logit E^ℓ | Min | **87.93**±1.01 | **91.24**±0.80 | 51.73±1.32 | 42.99±5.68 | 97.01±0.43 | **99.86**±0.16 | **84.53**±0.87 | 49.29±1.46 | 48.52±1.78 | **72.57**±22.36 |
| Spilled ΔE | Min | *79.04*±1.78 | *80.83*±1.87 | 43.22±1.67 | **74.36**±5.54 | **99.97**±0.08 | 61.97±2.81 | *78.54*±1.57 | 52.11±2.58 | 48.21±1.62 | *68.69*±17.48 |
| Spilled ΔE_s | Min | 77.75±1.52 | 79.44±2.05 | 43.39±1.82 | 72.87±6.10 | **99.97**±0.08 | 61.56±2.95 | 77.55±1.62 | *52.34*±2.57 | 48.17±1.62 | 68.12±17.15 |
| Marginal E^m | Max | 78.00±1.30 | 76.90±1.09 | 48.29±1.16 | 68.77±8.33 | 10.93±1.42 | *80.70*±1.98 | 67.49±1.69 | 51.91±2.32 | 51.28±2.47 | 59.36±20.69 |
| Marginal E^m | Min | 58.39±2.79 | 59.20±1.95 | 51.71±1.16 | 34.13±8.78 | 97.42±0.51 | 50.37±2.43 | 69.88±1.40 | 49.05±2.20 | 49.00±2.30 | 57.68±16.75 |
| Logit E^ℓ | Max | 53.47±2.13 | 49.02±1.79 | 48.27±1.32 | 57.38±6.09 | 91.76±0.91 | 57.42±1.43 | 52.77±2.58 | 50.74±1.51 | 51.17±1.83 | 56.89±12.70 |
| Logit E^ℓ | ALT | 43.56±1.95 | 39.74±1.73 | 48.27±1.32 | 57.41±6.06 | 91.71±0.94 | 43.11±1.57 | 43.62±2.57 | 50.74±1.51 | 51.17±1.83 | 52.15±14.88 |
| Logit E^ℓ | Last Token | 43.56±1.95 | 39.74±1.73 | 48.27±1.32 | 57.41±6.06 | 91.71±0.94 | 43.11±1.57 | 43.62±2.57 | 50.74±1.51 | 51.17±1.83 | 52.15±14.88 |
| Marginal E^m | ALT | 61.59±1.88 | 58.64±1.60 | 48.29±1.16 | 67.93±9.32 | 10.75±1.44 | 61.39±1.80 | 49.73±1.45 | 51.19±2.59 | 51.44±2.50 | 51.22±15.61 |
| Marginal E^m | Last Token | 61.59±1.88 | 58.64±1.60 | 48.29±1.16 | 67.93±9.32 | 10.75±1.44 | 61.39±1.80 | 49.73±1.45 | 51.19±2.59 | 51.44±2.50 | 51.22±15.61 |
| Marginal E^m | Mean | 58.27±2.50 | 58.64±1.58 | 48.29±1.16 | 68.32±8.35 | 6.12±0.70 | 66.55±3.22 | 45.67±1.38 | 51.80±2.29 | 51.29±2.46 | 50.55±17.33 |
| | | | | | Mistral-Instruct | | | | | | |
| Orgad et al. (2025) | Mean | 64.78±10.56 | 56.78±7.95 | **82.67**±11.63 | 68.78±11.43 | 64.22±12.12 | 64.89±11.55 | 65.44±12.10 | **61.00**±12.23 | **61.44**±11.31 | 65.56±6.84 |
| Spilled ΔE | Min | **91.12**±1.10 | 97.47±0.78 | 59.77±2.57 | 66.63±3.46 | 95.95±0.83 | **94.99**±0.93 | **91.75**±1.01 | 50.74±3.15 | 49.00±1.92 | **77.49**±19.42 |
| Marginal E^m | Min | 87.58±1.35 | **97.94**±0.62 | 18.67±2.27 | 67.58±3.37 | **97.96**±0.55 | 84.90±1.37 | *87.75*±1.73 | 49.19±3.97 | 48.49±1.86 | *71.12*±25.68 |
| Logit E^ℓ | Max | 77.24±1.66 | 83.84±1.66 | 22.28±2.54 | 57.67±3.29 | 78.98±1.58 | 76.89±1.49 | 80.35±1.88 | 45.53±2.60 | 48.17±1.97 | 63.44±19.99 |
| Marginal E^m | Max | 64.63±1.97 | 33.42±1.90 | *81.33*±2.32 | 26.52±2.28 | 17.62±1.20 | *86.60*±1.20 | 56.41±1.44 | 51.14±1.71 | | 53.68±22.53 |
| Logit E^ℓ | Last Token | 55.77±2.38 | 71.26±2.28 | 22.28±2.54 | **71.21**±2.42 | 47.78±2.26 | 42.93±1.96 | 58.36±3.52 | 45.65±2.94 | 48.30±2.04 | 51.50±14.26 |
| Logit E^ℓ | ALT | 55.77±2.38 | 71.26±2.28 | 22.28±2.54 | **71.21**±2.42 | 47.78±2.26 | 42.93±1.96 | 58.36±3.52 | 45.65±2.94 | 48.30±2.04 | 51.50±14.26 |
| | | | | | Mistral | | | | | | |
| Orgad et al. (2025) | Mean | 61.78±9.27 | 57.44±6.95 | **76.22**±12.82 | 65.78±15.27 | 56.67±11.83 | 64.22±8.91 | 64.33±10.40 | 58.00±12.29 | **54.56**±4.36 | 62.11±6.21 |
| Marginal E^m | Min | **87.52**±1.31 | **90.91**±1.58 | 54.69±2.49 | **86.21**±1.96 | **98.80**±0.35 | 94.41±0.62 | 83.66±2.16 | 52.15±1.74 | 46.37±2.02 | **77.19**±19.05 |
| Marginal E^m | Max | 83.57±1.13 | 86.83±1.70 | 45.31±2.49 | 62.26±4.29 | 96.03±0.83 | **99.27**±0.24 | **92.26**±1.31 | 51.31±3.35 | *54.49*±2.48 | 74.59±19.91 |
| Spilled ΔE | Min | *84.24*±1.18 | 83.74±1.41 | 57.43±2.99 | 78.26±2.93 | 96.69±0.62 | 84.47±1.17 | 81.27±1.83 | 50.62±1.72 | 48.72±1.75 | 73.94±16.18 |
| Spilled ΔE | Max | 61.50±1.88 | 63.60±1.68 | 42.57±2.99 | 76.27±3.42 | 47.01±2.48 | 81.84±1.60 | 68.07±1.30 | **58.71**±3.69 | 51.13±1.87 | 61.19±12.30 |
| Spilled ΔE_s | Max | 60.54±1.81 | 60.18±1.84 | 43.47±2.76 | 71.93±3.62 | 45.94±2.40 | 78.84±1.53 | 67.92±1.32 | 57.24±3.72 | 51.88±1.90 | 59.77±11.08 |

Table 5: Hallucination detection performance, in terms of AuROC, across nine benchmarks and different LLMs. We measure the generalization across all tasks by computing the average.

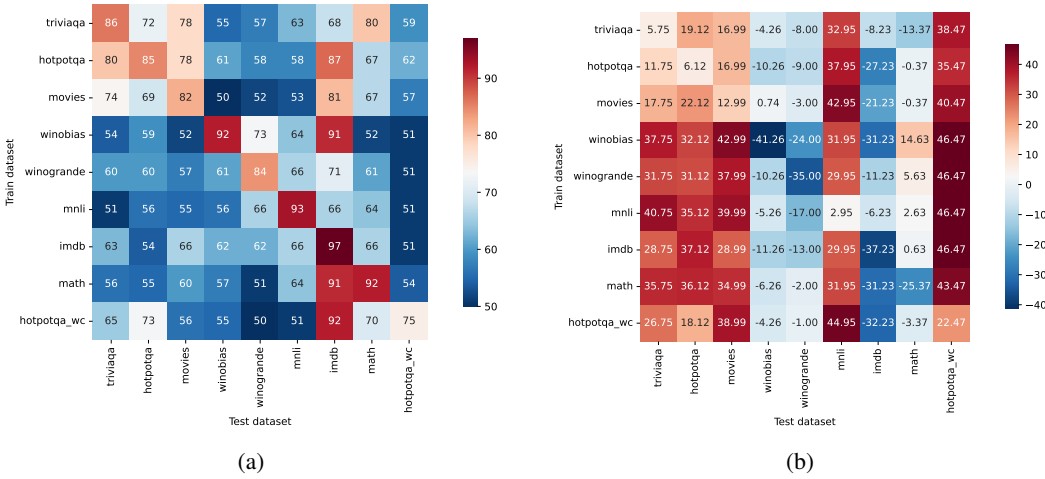

(a)                    (b)

Figure 10: Fig. 10a presents the cross-dataset performance of the method proposed by Orgad et al. (2025) using Mistral-Instruct. Fig. 10b depicts the performance difference between their method and our Spilled ΔE with Min pooling. Positive values indicate cases where Spilled ΔE outperforms the method of Orgad et al. (2025). All numbers are computed as percentages.

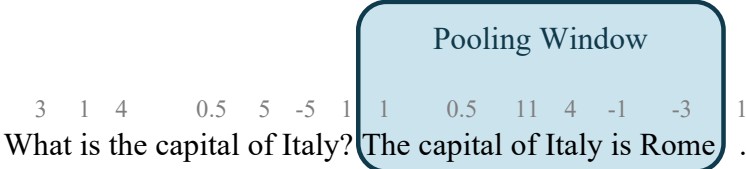

Figure 11: Example of the Pooling Window

## D.3 ADDITIONAL RESULTS FOR CROSS-TESTING WITH REAL WORLD BENCHMARKS

Table 5 shows how our method compares with the baseline methods, Orgad et al. (2025) and Logit $E^\ell$. This table was obtained by using various pooling methods in the pooling window from which we measure possible hallucinations. More details below based on the example in Fig. 11:

⋄ **Min**: minimum energy value in the pooling window. Energy Measured: $-3$

⋄ **Max**: maximum energy value in the pooling window. Energy Measured: 11

⋄ **Mean**: mean of all the energies in the pooling window. Energy Measured: 2.08

⋄ **Last Token**: energy of the last token in the pooling window. Energy Measured: $-3$

⋄ **After Last Token**: energy of the first token after the pooling method. Energy Measured: 1

### D.3.1 SUCCESS CASES

Question: ``Which planet is known as the Red Planet?''

> **Logits:** The Red Planet is Mars . ✓
>
> **Ours:** The Red Planet is Mars . ✓

> **Logits:** The Red Planet is Jupiter . ✗
>
> **Ours:** The Red Planet is Jupiter . ✗

Question: ``What is the largest mammal in the world?''

> **Logits:** The largest mamm al in the world is the Blue Whale ✓
>
> **Ours:** The largest mamm al in the world is the Blue Whale ✓

> **Logits:** The largest mamm al in the world is the House Cat . ✗
>
> **Ours:** The largest mamm al in the world is the House Cat . ✗

Question: ``Who painted the Mona Lisa?''

> **Logits:** The Mona Lisa was painted by Leonardo da Vinci . ✓
>
> **Ours:** The Mona Lisa was painted by Leonardo da Vinci . ✓

> **Logits:** The Mona Lisa was painted by Pablo Esc obar . ✗
>
> **Ours:** The Mona Lisa was painted by Pablo Esc obar . ✗

Question: ``What gas do plants breathe in for photosynthesis?''

> **Logits:** They breathe in carbon dioxide ✓
> **Ours:** They breathe in carbon dioxide ✓

> **Logits:** They breathe in oxygen ✗
> **Ours:** They breathe in oxygen ✗

Question: ``In which continent is Egypt Located?''

> **Logits:** Egypt is located in Africa ✓
> **Ours:** Egypt is located in Africa ✓

> **Logits:** Egypt is located in Europe ✗
> **Ours:** Egypt is located in Europe ✗

Question: ``What is the fastest land animal?''

> **Logits:** The fastest land animal is the che et ah ✓
> **Ours:** The fastest land animal is the che et ah ✓

> **Logits:** The fastest land animal is the lion ✗
> **Ours:** The fastest land animal is the lion ✗

Question: ``What is the hardest natural substance on Earth?''

> **Logits:** The hardest natural substance is diamond ✓
> **Ours:** The hardest natural substance is diamond ✓

> **Logits:** The hardest natural substance is gold ✗
> **Ours:** The hardest natural substance is gold ✗

Question: ``Which ocean is the largest?''

> **Logits:** The largest ocean is the Pacific Ocean ✓
> **Ours:** The largest ocean is the Pacific Ocean ✓

> **Logits:** The largest ocean is the Indian Ocean ✗
> **Ours:** The largest ocean is the Indian Ocean ✗

### D.3.2 FAILURE CASES

Question: ``Who was the first person to walk on the moon?''

> **Logits:** Neil Armstrong ✓
> **Ours:** Neil Armstrong ✓

> **Logits:** Buzz Ald rin ✗
> **Ours:** Buzz Ald rin ✗

Reasoning: ``If there are 3 cars and each car has 4 wheels , how many wheels are there in total? ''

> **Logits:** Each car has 4 wheels . So , for 3 cars , the total number of wheels is 3 x 4 = 12 wheels . ✓
> **Ours:** Each car has 4 wheels . So , for 3 cars , the total number of wheels is 3 x 4 = 12 wheels . ✓

> **Logits:** Each car has 8 wheels . So , for 3 cars , the total number of wheels is 3 x 8 = 14 wheels . ✗
> **Ours:** Each car has 8 wheels . So , for 3 cars , the total number of wheels is 3 x 8 = 14 wheels . ✗

Reasoning: ``What is the square root of 64?''

> **Logits:** The square root of 64 is 8 ✓
> **Ours:** The square root of 64 is 8 ✓

> **Logits:** The square root of 64 is 10 ✗
> **Ours:** The square root of 64 is 10 ✗

Question: ``What blood type is known as the universal donor?''

> **Logits:** O negative ✓
> **Ours:** O negative ✓

> **Logits:** AB positive ✗
> **Ours:** AB positive ✗

