# OpenReview forum: "Spilled Energy in Large Language Models"
_ICLR.cc/2026/Conference — ICLR 2026 Poster_

### Official Review · Reviewer_LpMe · 2025-10-26

**Soundness:** 3
**Presentation:** 3
**Contribution:** 3
**Rating:** 6
**Confidence:** 4

**Summary:**

This paper proposes a new method for detecting hallucinations in language models by interpreting the model's sequential predictions as quotients of two different energy-based models (EBMs) denoted $E_\theta^{\ell}$ and $E_\theta^m$. The paper defines the spilled energy as the difference between the two energies: $\Delta E_\theta(\mathbf{x}_{0:i}) = E_\theta^\ell(\mathbf{x}_{0:i}) - E_\theta^m(\mathbf{x}_{0:i})$.

The paper then proposes several measures for hallucination detection, including:
- marginal energy $E_\theta^m$,
- spilled energy $\Delta E_\theta$, and
- scaled spilled energy $\Delta E_s := |E_\theta^m| \Delta E_\theta$.

These metrics are evaluated in two settings. First, a synthetic setup, involving the detection of increasingly smaller numerical errors introduced in arithmetic computations of 13-digit integers. Spilled energy ($\Delta E_\theta$) gets strictly superior ROC curves (better pointwise, not just in AUC) than baselines across Llama-3-8B (both instruct and non-instruct versions), Qwen-3-8B, and Mistral-7B-Instruct.

Second, the proposed methods are tested in detecting errors in reasoning tasks across several benchmarks. To do so, the *exact answer tokens* are detected, and the different methods are computed across them using different pooling methods. The different methods are compared against logit energy and [1] using AuROC scores.

---

[1] Orgad, H., Toker, M., Gekhman, Z., Reichart, R., Szpektor, I., Kotek, H. and Belinkov, Y., 2024. Llms know more than they show: On the intrinsic representation of llm hallucinations. arXiv preprint arXiv:2410.02707.

**Strengths:**

- **(S1)** The proposed methods are efficient, simple, generic, easy to implement, can be applied to any language model given access to its logits (gray-box access), and extend previous hallucination detection methods.

- **(S2)** The proposed methods are evaluated on a wide range of models and tasks, and show strong performance. The synthetic setup is well-designed, and the reasoning tasks are representative of the kind of tasks where hallucinations are likely to occur in real-world applications.

- **(S3)** The entire derivation of spilled energy is presented in the main text and is not hidden in the appendix. The derivation is clear and easy to follow (if a bit cumbersome).

**Weaknesses:**

- **(W1)** The paper does not provide a theoretical justification for why spilled energy (or the other proposed methods) is a good indicator of hallucinations. It is just an empirical observation. Moreover, the intuitive explanation is limited to:
  > ...the two energies, not interacting at the same step but at steps $i$ and $ i-1$, should be equal, but they are measured in the LLM at different generation steps and from different components.

  and
  > Since both terms on the right side should be equal to $E_\theta (\mathbf{x}_{i:1})$, delta values should always be zero when we are correctly modeling the energy at timestep $i$.

  Which are unclear to me (see question 1).

- **(W2)** While the authors mention that using the exact answer is critical for deployment, the paper is vague on the exact method used to detect these exact answer tokens. The only mention I found is in Section 4.2:
  > We identify this span by prompting the LLM for a brief answer.

- **(W3)** The derivation of spilled energy is more cumbersome and complicated than it should be. E.g., equations (1) and (2) are redundant (one of them suffices), and equations (3)-(5) can be united into one. This is a minor weakness.

**Questions:**

- **(Q1)** Why do the authors claim that:
>  ... delta values should always be zero when we are correctly modeling the energy at timestep $i$

- **(Q2)** How exactly are the exact answer tokens detected?

- **(Q3)** When comparing with [1] in Table 1, are the probes retrained on each dataset? If not, which datasets are they trained on?

Given a strong answer to **Q1**, addressing **W1**, I would be open to increasing the score.

---

[1] Orgad, H., Toker, M., Gekhman, Z., Reichart, R., Szpektor, I., Kotek, H. and Belinkov, Y., 2024. Llms know more than they show: On the intrinsic representation of llm hallucinations. arXiv preprint arXiv:2410.02707.

---

> ### Author Response · Authors · 2025-11-23
> **Answer to Reviewer LpMe (1/4)**
>
> Thank you for the thoughtful and constructive feedback. We are pleased that several important strengths of our work have been highlighted, many of which were independently emphasized by the other reviewers as well:
>
> - the acknowledgment that our derivation of spilled energy is **clear, explicit, and easy to follow**, noting that the full derivation appears in the main text (not hidden in the appendix), and this aligns with `zjL6` and `qt4T`, who both described the paper as well-written and mathematically sound;
> - the recognition that our proposed metrics are **efficient, simple, generic, and easy to implement**, a theme echoed by `zjL6` (“simple and training free”) and `Xm2r` (“lightweight and applicable to most LLMs”), emphasizing the practicality of our approach;
> - the positive assessment of our **broad and rigorous experimental evaluation**, highlighting that the synthetic setup is well-designed and the reasoning benchmarks are representative, a point reinforced by `Xm2r` (“results on both synthetic and real-world datasets validate generalization”) and `zjL6` (“strong cross-dataset generalization across LLaMa-3, Mistral, Qwen-3”);
> - the appreciation of our method’s **generality across model families and datasets**, reflected in your remark that we evaluate on “a wide range of models and tasks,” consistent with `Xm2r` (“validated across model family”) and `zjL6` (“across synthetic and real-world datasets”);
> - the recognition of the **strength and relevance of our contributions**, noting that our measures extend prior hallucination detection methods while remaining efficient and broadly applicable, aligning with similar sentiments from `zjL6` and `Xm2r`.
>
> We appreciate your assessment, and we are encouraged by the strong convergence across reviewers regarding the clarity, simplicity, empirical rigor, and general applicability of our approach.

---

> ### Author Response · Authors · 2025-11-23
> **Answer to Reviewer LpMe (2/4)**
>
> ### (W1, Q1) Theoretical justification and why delta needs to be zero
>
> The theoretical justification might not seem very intuitive, but it is indeed well motivated and supported by a logical rationale: the two quantities, $E\_\theta^\ell(\mathbf{x}\_{0:i})$ and $E\_\theta^m(\mathbf{x}\_{0:i})$ of Eq. (8) in the paper, represent the **same** quantity in the theoretical framework of EBMs applied to classifiers used for sequence modeling; thus, they should cancel each other out. **Canceling out implies that the difference is zero.** This is not implemented as regularization at training time, so this represents a degree of freedom for the model, resulting in spilled energy.
>
> A more thorough explanation follows.
>
> Let us assume a sequence of three tokens  $\mathbf{x}_2, \mathbf{x}_1, \mathbf{x}_0$ . If we do language modeling with autoregression, minimizing the negative log-likelihood, we have:
>
> $$
> -\log p(\mathbf{x}\_2, \mathbf{x}\_1, \mathbf{x}\_0) = -\log \underbrace{p(\mathbf{x}\_2|\mathbf{x}\_1,\mathbf{x}_0)}\_{\text{step 2}}p(\mathbf{x}\_1|\mathbf{x}\_0)p(\mathbf{x}\_0)
> $$
>
> Now, every conditional probability on the right side is implemented with a transformer ending in a softmax discriminative classifier. Equations (3) and (5) in our paper allow us to re-interpret:
>
> $$
> \text{\textbf{step 2:}}\quad -\log p(\mathbf{x}_2|\mathbf{x}_1,\mathbf{x}_0) = -\log \frac{p(\mathbf{x}_2,\mathbf{x}_1,\mathbf{x}_0)}{p(\mathbf{x}_1,\mathbf{x}_0)} = -\log \Bigg[ \frac{ \exp\big(\theta(\mathbf{x}_1,\mathbf{x}_0)[id(\mathbf{x}_2)]\big)}{\sum_k^V\exp\big(\theta(\mathbf{x}_1,\mathbf{x}_0)[k]\big)} \Bigg] = E^l(\mathbf{x}_2,\mathbf{x}_1,\mathbf{x}_0)-E^m(\mathbf{x}_1,\mathbf{x}_0).
> $$
>
> In other words, we reinterpret:
>
> - the numerator $p(\mathbf{x}_2,\mathbf{x}_1,\mathbf{x}_0)$ as the energy $E^l(\mathbf{x}_2,\mathbf{x}_1,\mathbf{x}_0)$, which is the **logit (l)** of the softmax at timestep 2;
> - The denominator as the energy $E^m(\mathbf{x}_1,\mathbf{x}_0)$ obtained with the **marginalization ($**m$**)** across the vocabulary $V$. This value can be read  “read” simply by taking the denominator of the softmax at timestep 2. Please remember this term.
>
> It is better to indicate them as energies (since they are not probabilities), and given their logarithmic properties, we obtain a difference. We use the notation $l$ for logits and $m$ for marginalization.
>
> Now, **when we go across steps and we connect two-time steps, this is where the magic happens:**
>
> $\text{\textbf{step 1:}}\quad-\log p(\mathbf{x}_1|\mathbf{x}_0) = -\log \frac{p(\mathbf{x}_1,\mathbf{x}_0)}{p(\mathbf{x}_0)} = E^l(\mathbf{x}_1,\mathbf{x}_0)-E^m(\mathbf{x}_0).$
>
> We see that at timestep 1, the value $E^l(\mathbf{x}_1,\mathbf{x}_0)$ **appears again, but measured at the logit level.**
>
> In other words, across the time-steps 2 and 1, the quantity $E(\mathbf{x}_1,\mathbf{x}_0)$ is measured twice:
>
> - at timestep 2, as the marginalization
> - at timestep 1, as the logit.
>
> In the architecture or in the loss, there is no mechanism that is forcing this to be the **same, but they should be equal, given the language modeling objective.** This is the same as saying that in Equation (2) of our paper, the probabilities across time steps need to be simplified as we show.
>
> In other words, this:
>
> $p(\mathbf{x}_2, \mathbf{x}_1, \mathbf{x}_0) = p(\mathbf{x}_2|\mathbf{x}_1,\mathbf{x}_0)p(\mathbf{x}_1|\mathbf{x}_0)p(\mathbf{x}_0)$
>
> implies:
>
> $$E(\mathbf{x}\_2,\mathbf{x}\_1,\mathbf{x}\_0) = E^l(\mathbf{x}\_2,\mathbf{x}\_1,\mathbf{x}_0) \underbrace{- E^m(\mathbf{x}\_1,\mathbf{x}\_0) + E^l(\mathbf{x}\_1,\mathbf{x}\_0)}\_{\text{should be zero}} \underbrace{- E^m(\mathbf{x}\_0) + E^l(\mathbf{x}\_0)}\_{\text{should be zero}}$$
>
> To model the energy of a sequence $E^l(\mathbf{x}_2,\mathbf{x}_1,\mathbf{x}_0)$ correctly, then:
>
> - $- E^m(\mathbf{x}_1,\mathbf{x}_0) + E^l(\mathbf{x}_1,\mathbf{x}_0) = 0$ (spilled energy at timestep 2 if non-zero)
> - $-E^m(\mathbf{x}_0) + E^l(\mathbf{x}_0) = 0$ (spilled energy at timestep 1 if non-zero)
>
> so that $E(\mathbf{x}_2,\mathbf{x}_1,\mathbf{x}_0) = E^l(\mathbf{x}_2,\mathbf{x}_1,\mathbf{x}_0)$.
>
> Our rationale for the spilled energy going to zero for well-modeled sentences is also supported by the histogram in Fig. 2(c-d): in practice, spilled energy is never *exactly* zero, but there is a point in the fact that **correct answers are most of the time much closer to zero than wrong answers.** This is visible in:
>
> - Fig. 2(c): the green histogram of spilled energy for the correct answer is centered at zero with some variability; the more we make the arithmetic difficult, the more the spilled energy moves to the right, far from zero.
> - Fig. 2(d): the same holds in real-world datasets; the histogram of correct answers is shifted more towards zero compared to the wrong histograms.
>
> In *practice*, the **less energy spills, the less hallucination we experience**. This is the point of our paper.
>
> For additional information, please see our reply to **(W4)** reviewer `Xm2r`.

---

> ### Author Response · Authors · 2025-11-23
> **Answer to Reviewer LpMe (3/4)**
>
> ### (W2, Q2) Exact Answer Token Detection
> > How exactly are the exact answer tokens detected?
>
> We appreciate the reviewer raising this crucial question regarding the deployment details of our method. We recognize that the brief mention in the main text may have led to an incomplete understanding of this procedure.
>
> The precise methodology for **detecting exact answer tokens** is now fully documented in **Appendix B.1**.
>
> **Summary of the Detection Method**
>
> Our primary focus was on isolating the analytical gains from using the exact answer tokens, rather than on optimizing the extraction tool itself. The extraction method, which is the contribution of Orgad et al.(2025), uses a combination of strategies:
>
> 1. **Heuristics for Closed Sets:** For tasks with a finite set of possible answers (e.g., Winobias, MNLI, IMDB), we primarily rely on simple **heuristic methods** to identify the answer span.
> 2. **Instruction-Tuned LLM for Open-Ended Tasks:** For open-ended scenarios like factual question answering (e.g., TriviaQA) or Reasoning (e.g. Math), we utilize an **instruction-tuned LLM** (Mistral-7B-Instruct in our experiments) to extract the precise answer substring.
>     - This extraction model is prompted with the question and the full LLM long answer to output only the "relevant tokens".
>     - To ensure robustness, the extraction process includes steps to verify the extracted text is a substring of the generated answer and is **rerun up to five times** if necessary.
> 3. **Token Identification:** Once the exact answer substring is found, we perform a simple search to locate its token boundaries within the long response.
>
> We would like to clarify that **Table 3** in Appendix A.2 of Orgad et al. (2025) demonstrates the high success rate (e.g., $0.95-0.99$) of this process, even when using the LLM itself to extract its own answer token, which supports the viability of this step.
>
> We confirm that we have ensured this detailed explanation remains clear in the revised version of our paper. We sincerely thank the reviewer for helping us improve the clarity of the paper's structure.
>
> ### (W3) Redundancy in equations
> > The derivation of spilled energy is more cumbersome and complicated than it should be. E.g., equations (1) and (2) are redundant (one of them suffices), and equations (3)-(5) can be united into one. This is a minor weakness.
>
> We agree that the sequence of equations may appear more verbose than strictly necessary, but our intention was to provide a rigorous, step-by-step connection between established LLM methodology and the Energy-Based Model (EBM) framework.
>
> - **Regarding Eq. (1) and Eq. (2):** These equations serve distinct conceptual roles. **Equation (1)** establishes the standard autoregressive factorization of the joint probability $p(x_{i:1})$ and explicitly highlights how LLMs achieve *generative modeling* by recursively applying a *discriminative classifier* (the conditional $p_{\theta}(x_i|x_{i-1:1})$). **Equation (2)** then builds on this by conceptually expanding the product into a ratio of joint probabilities, providing the immediate mathematical intuition of the cancellation property that *should* hold true, but which our work exploits as a source of "spilled energy" when it fails in practice.
> - **Regarding Equations (3)-(5):** We maintain that decomposing the EBM reinterpretation into these steps is crucial for clarity. **Equation (3)** first introduces the standard softmax expression for the conditional probability, linking the logit vector $\theta(x_{i-1:1})$ directly to the probability ratio $\frac{p_{\theta}(x_{i:1})}{p_{\theta}(x_{i-1:1})}$. **Equation (4)** formally restates this relationship using the EBM framework's log-likelihood form. Finally, **Equations (5) and (6)** provide the literal, term-by-term mapping from the conditional log-likelihood (Eq. 4) back to the logits (Eq. 3), explicitly defining our core quantities: the sampled logit energy, $E_{\theta}^l$, and the marginal energy, $E_{\theta}^m$. Uniting them would obscure this essential mapping of internal LLM components to the energy terms.
>
> We believe this detailed scaffolding is necessary to ensure the mathematical foundation of our proposed $\Delta E_{\theta}(x_{i:1})$ metric is fully transparent and principled, particularly when reinterpreting the conventional LLM architecture through the EBM lens.

---

> ### Author Response · Authors · 2025-11-23
> **Answer to Reviewer LpMe (4/4)**
>
> ### (Q3) Probing Classifier Training Strategy for Cross-Dataset Results
> > When comparing with [1] in Table 1, are the probes retrained on each dataset? If not, which datasets are they trained on?
>
> The results reported for the **Orgad et al. (2025) Probing Classifiers** in our paper's Table 1 (cross-dataset results, Section 5.2) *do not* represent a model trained and tested on the same dataset. Instead, they represent performance under a stringent cross-dataset generalization (transfer) regime.
>
> **Training and Testing Methodology for Table 1**
>
> To measure generalization, the performance value reported for any given test dataset $D_{test}$ (e.g., TriviaQA) is the average performance across classifiers trained on *each* of the datasets $D_{train} \in \{D\}$ and then evaluated on $D_{test}$.
>
> Specifically, if $\mathcal{D}$ is the set of nine datasets:
>
> $\text{Performance}(D\_{test}) = \text{Average}\_{D\_{train} \in \mathcal{D}} \left( \text{AuROC}(\text{Probe trained on } D\_{train}, \text{tested on } D\_{test}) \right)$
>
> For example, the final AuROC value reported for **TriviaQA** is the average test score from the probing classifiers trained individually on:
> $\{\text{TriviaQA}, \text{HotpotQA}, \text{HotpotQA-WC}, \text{IMDB}, \text{Math}, \text{MNLI}, \text{Movies}, \text{Winobias}, \text{Winogrande}\}$
> and subsequently tested on $\text{TriviaQA}$.
>
> This metric is designed to show that, unlike our training-free methods (like $\Delta E_{\theta}$), the probe classifiers **struggle significantly with cross-dataset transfer**, reflecting a non-universal encoding of truthfulness. We note that the reported performance for the Orgad et al.(2025) probes drops sharply under this cross-dataset evaluation.
>
>
> ---
> Moreover, for more in-depth information on the performance of each dataset, please refer to the **revised manuscript, specifically Fig. 4**.
>
> **Fig. 4** is a “confusion” matrix of the probe performance within and across datasets (Fig. 4(a)), along with a comparison with our spilled energy method (Fig. 4(b)). This matrix, displayed as a heatmap, captures both in-distribution (diagonal elements) and out-of-distribution performance (off-diagonal elements).
>
> In **Fig. 4(b)**, we clearly see that the spilled energy greatly improves over the baseline, both off-diagonally (cross-dataset) and sometimes even within the diagonal (within the same dataset).
>
> Figures 8, 9, and 10 for other LLMs are included in the Appendix.
>
> Additional information is in the **answer to Reviewer `qt4T`(6/6).**

---

> ### Comment · Area_Chair_Mbgu · 2025-11-28
>
> Dear Reviewer, the discussion period is about to close. We kindly ask you to participate in the discussion or update your score based on the authors' rebuttal before the deadline. Thank you for your time and valuable contribution!

---

### Official Review · Reviewer_Xm2r · 2025-10-27

**Soundness:** 3
**Presentation:** 3
**Contribution:** 3
**Rating:** 4
**Confidence:** 3

**Summary:**

This paper proposes a training-free method for detecting hallucinations in LLM. The core idea is to reinterpret the final softmax layer of an LLM as an Energy-Based Model (EBM). The paper define a variable named spilled energy that quantifies the differences between two energy values across consecutive generation steps that should ideally be equal. The paper has a hypothesis that a big spilled energy is correlate with hallucination in model generation. The authors test this hypothesis on both synthetic task and real-world benchmarks. The results show that spilled energy is a strong signal for detecting hallucination. Their method outperforms baseline method, although there could still be false positive cases.

**Strengths:**

1. The paper formulate the LLM's generation procedure as energy based model, which enable the following definition of "spilled energy" for hallucination detection.
2. The method proposed in the paper is a training-free method which make it lightweight and applicable to most LLM for hallucination detection. Empirically, the paper show results on both sythentic and realworld dataset, which validate that the method may have generalization across different domains. While other works (non-training free) show worse performance under new datasets, the paper’s method show robustness in performance.
3. The experiments in the paper are solide. The author provide results on sythentic dataset, which show the metric's ability to separate correct from incorrect answers. The results on realworld datasets shows the method's valid on a wide range of domains. The results are validated across model family as well.
4. The ablation section also consolidate the findings. The authors demonstrate the critical importance of localizing the "exact answer" tokens, showing that doing so provides a good performance gain.

**Weaknesses:**

1. The method needs to first identify the specific token span constituting the "exact answer". The paper implement this by "prompting the LLM for a brief answer”. My concern is that what if the LLM's "brief answer" is itself a hallucinated? What if the “exact answer” is wrongly identified?
2. The current evaluation focuses on tasks with answer that can be localized to a short range(words). My question is that how the method perform on more subtle hallucinations, such as a mutilple incorrect words/phrases in a long paragraph or complex factual error that is not contained in a single noun phrase.
3. In limitation, the author acknowledge that the method can produce false positives on tokens that are not semantically informative, such as punctuation. While this might be mitigated this with answer localization, it suggests the raw signal is noisy and may not be a pure measure of semantic or factual correctness. Can we do better to reduce the false positive?
4. If energy gets “spilled”, it is a sign the model’s prediction doesn’t align well with its internal probability distribution. Can you give more explanations why we can use this to detect hallucination? I am not very convinced by current explaination in paper.
5. Each time, we need to inference the model twice, the extra latency make the method impractical

**Questions:**

see weakness

---

> ### Author Response · Authors · 2025-11-23
> **Answer to Reviewer zjL6 (1/5)**
>
> Thank you for the thoughtful and constructive feedback. We are pleased that several important strengths of our work were highlighted, many of which were independently emphasized by the other reviewers as well:
>
> - The recognition that our method is **training-free, lightweight, and broadly applicable to most LLMs**, a point made explicitly by `Xm2r` (“training-free… lightweight and applicable to most LLM”), and mentioned by `zjL6` (“simple and training free”) and acknowledged positively by `qt4T`;
> - The appreciation of our **EBM-based formulation**, with `Xm2r` noting that the paper “formulate[s] the LLM’s generation procedure as energy based model,” aligning with `zjL6` who described this idea as “novel and very interesting,” and with `LpMe` who highlighted the simplicity and generality of the approach;
> - The acknowledgement that our **experiments are strong, thorough, and span both synthetic and real-world datasets**, emphasized by `Xm2r` (“results […] validate that the method may have generalization across different domains”), and similarly noted by `zjL6` (“strong cross-dataset generalization”) and `LpMe` (“synthetic setup is well-designed; real-world benchmarks are representative”);
> - the validation that our method **generalizes across model families**, as noted directly by `Xm2r` (“validated across model family”), and reinforced by `zjL6` (“across LLaMa-3, Mistral, Qwen-3”) and `LpMe` (“evaluated on a wide range of models and tasks”);
>
> ---
>
> We now address the remaining remarks, with the hope that they will enhance the evaluation of our work.
>
> ### (W1) Possible hallucinations in Exact Answer extractions
>
> > The method needs to first identify the specific token span constituting the "exact answer". The paper implement this by "prompting the LLM for a brief answer”. My concern is that what if the LLM's "brief answer" is itself a hallucinated? What if the “exact answer” is wrongly identified?
>
> This is a good remark and was also our thought when we read for the first time the paper by Orgad *et al*, published at the previous ICLR 2025. We point out that the exact answer extraction is not our contribution, **and** we only confirm the finding of Orgad *et al.,* which provides solid ground for our analysis of spilled energy.
>
> We want to highlight that **exact token performance is very stable and reliable:** We confirm the finding of Orgad *et al* that Exact token performance is also very stable and reliable, at least on the 9 benchmarks that we processed. As shown in Table 3 of Appendix A.2 in Orgad et al.[1], Answer extraction has a very high success rate, ranging from 0.95 (Llama3-Instruct-8B) to 0.99 (Mistral-7B).
>
> Since the extracted answer must then be located within the original generated answer, if any failure occurs, we retry extraction up to five times. We achieved the same accuracy for exact answers in our implementation, as we started from the code of Orgad et al.
>
> The extracted answer must then be located within the original generated answer, which acts as an additional safety mechanism. We achieved similar accuracy for answer localization, as we started from the code implementation of Orgad et al.
>
> We report in the following table, the extraction success rate across the full datasets using Mistral-Instruct-v0.2, which has also been included in section B.1 of the appendix:
>
> | Dataset | Success Rate (%) |
> | --- | --- |
> | TriviaQA | 90.29 |
> | HotpotQA | 87.37 |
> | Movies | 93.61 |
> | Math | 87.59 |
> | HotpotQA-WC | 92.38 |
>
> Note that some datasets have been excluded (IMDB, Winobias, Winogrande, MNLI), since they have a finite set of possible answers which can be used to easily locate the exact answer within the model’s generation.
>
> This task of localization can be in the future further optimized either using Small Language Models (SLMs) or training a small sequential classifier, not to detect if there is an hallucinations over activations, but on the sequence of energies to detect where are the exact tokens: the supervision for this small recurrent classifier can be given by distilling the current method to find the exact token.
>
> Similar to Orgad *et al*, in the main text, we intentionally keep the extraction step abstract, as **our goal is not to commit to a particular implementation but to study the benefits that arise once these locations are identified**. The point of our paper is to **demonstrate that we can achieve outstanding generalization performance with a training-free method by reinterpreting the LM chain as EBMs**.

---

> ### Author Response · Authors · 2025-11-23
> **Answer to Reviewer zjL6 (2/5)**
>
> ### (W2) Detecting hallucinations in long paragraphs or complex factual errors
>
> > The current evaluation focuses on tasks with answer that can be localized to a short range(words). My question is that how the method perform on more subtle hallucinations, such as a mutilple incorrect words/phrases in a long paragraph or complex factual error that is not contained in a single noun phrase.
>
> We agree that detecting subtle hallucinations spread across extended text, particularly those not reducible to a single key entity, presents a significant challenge for current methods.
> While our current evaluation's main focus are tasks amenable to **exact answer token localization** (e.g., factual and common-sense QA), we do explore reasoning through HotpotQA by not providing context, forcing models to reason about their own knowledge before providing an answer. We also explore more complex reasoning traces through the **Math dataset** experiments.
>
> Our method addresses long-form generation by focusing its measurement on the identified **answer window** or key tokens within the reasoning trace. Although our current $\Delta E_{\theta}$ framework does not track energy fluctuations for tokens generated *prior* to the final answer, our empirical results on arithmetic and complex QA tasks show that the chosen answer window **preserves crucial energy dynamics** highly correlated with the occurrence of a hallucination elsewhere in the generation process. This demonstrated robustness in detecting arithmetic and reasoning failures suggests a promising direction for extending this analysis to more diffuse and complex factual errors.
>
> We report results for the Math dataset to highlight the performance of our methods across different models in this challenging setting.
>
> ### Mistral-7B-v0.3
>
> | Method | Math |
> | --- | --- |
> | p-True | 45.86 ± 2.05 |
> | Logit | 57.21 ± 3.89 |
> | Orgad et al. | 65.78 ±15.27 |
> | Spilled Energy | 78.26 ± 2.93 |
> | Marginal Energy | **86.21 ± 1.96** |
>
> ### Mistral-7B-Instruct-v0.2
>
> | Method | Math |
> | --- | --- |
> | p-True | 51.63 ± 1.29 |
> | Logit | 57.67 ± 3.29 |
> | Orgad et al. | **68.78 ± 11.43** |
> | Spilled Energy | **66.63 ± 3.46** |
> | Marginal Energy | **67.58 ± 3.37** |
>
> ### Meta-Llama-3-8B-Instruct
>
> | Method | Math |
> | --- | --- |
> | p-True | 49.53 ± 2.17  |
> | Logit | 57.81 ± 3.82 |
> | Orgad et al. | 66.56 ± 17.04 |
> | Spilled Energy | 65.58 ± 3.02 |
> | Marginal Energy | **70.55 ± 2.43** |
>
> ### Meta-Llama-3-8B
>
> | Method | Math |
> | --- | --- |
> | p-True | 58.63 ± 1.26 |
> | Logit | 57.38 ± 6.09 |
> | Orgad et al. | 69.67 ± 15.07 |
> | Spilled Energy | **74.36 ± 5.54** |
> | Marginal Energy | 68.77 ± 8.33 |
>
> Given the high accuracy scores for detection, we believe Spilled Energy and Marginal Energy's performance is reliable even in these scenarios.

---

> ### Author Response · Authors · 2025-11-23
> **Answer to Reviewer zjL6 (3/5)**
>
> ### (W3) Mitigating False Positives in Spilled Energy
>
> > In limitation, the author acknowledge that the method can produce false positives on tokens that are not semantically informative, such as punctuation. While this might be mitigated this with answer localization, it suggests the raw signal is noisy and may not be a pure measure of semantic or factual correctness. Can we do better to reduce the false positive?
>
> We acknowledge the reviewer's excellent point: the observation of high $\Delta E_{\theta}$ on non-semantic tokens (like punctuation) indicates that the raw signal may not be a pure measure of semantic or factual correctness. While answer localization is our current primary mitigation strategy, we agree that the base signal noise warrants further attention.
>
> To directly address this, we believe implementing classic NLP techniques offers a promising, lightweight solution:
>
> - **Linguistic Filtering via POS Tagging:** We propose utilizing traditional tools, such as **Part-of-Speech (POS) tagging** and Named Entity Recognition (NER), to filter the tokens considered for measurement. This would allow us to prioritize calculating the pooled energy over tokens identified as **semantically informative** (e.g., nouns, verbs, named entities) while discarding contributions from purely functional tokens (e.g., punctuation, prepositions, determiners), which are often responsible for false positives. This technique could potentially localize the signal more effectively than simple answer span detection alone.
>
> Furthermore, future work could explore more advanced, intrinsic filtering methods:
>
> - **Attention Masking:** Since $\Delta E_{\theta}$ is based on logits derived from the transformer's output, it may be beneficial to analyze the attention patterns. Specifically, investigating if the **causal attention mask** over these non-semantic tokens reveals low-entropy distributions, which could signal tokens where the model is globally uncertain, leading to the observed energy spill.
>
> By incorporating linguistic features for post-processing or attention analysis for intrinsic filtering, we can aim to isolate the energy dynamics related to genuine factual uncertainty, thereby significantly improving the reliability of the $\Delta E_{\theta}$ signal.

---

> ### Author Response · Authors · 2025-11-23
> **Answer to Reviewer zjL6 (4/5)**
>
> ### (W4) Clarification on Spilled Energy’s link with Hallucinations
>
> > If energy gets “spilled”, it is a sign the model’s prediction doesn’t align well with its internal probability distribution. Can you give more explanations why we can use this to detect hallucination? I am not very convinced by current explaination in paper.
>
> Other researchers, for example, *Liu et al. (2020),* cited in our paper, found a more effective way to assess whether a classifier can be trusted in its predictions, not by evaluating its softmax probabilities but by reinterpreting the classifier as an EBM and measuring the energy instead. *Liu et al. (2020)* demonstrate that marginal energy is a stronger predictor for out-of-distribution data, aligns better with the probability density of the inputs, and is less susceptible to the overconfidence issue. This prior work motivates the use of energy instead of plain probabilities for LLMs, which is what the paper primarily uses to assess the credibility of LLMs.
>
> Our paper presents a similar concept, **but we apply it to the chain of probabilities that an LLM is trained on to perform language modeling (LM).**
>
> Consider Eq. (2) in our paper and the simplification that occurs between the two probabilities between step *i* and step *i-1*: that simplification occurs **because the probability in the denominator at step *i* is the same as the probability in the numerator at step i-1 in order to perform language modeling correctly.** We measure those inside and LLMs in terms of energy, and the **spilled energy is the amount by which they differ.**
>
> Let us assume a sequence of three tokens  $\mathbf{x}_2, \mathbf{x}_1, \mathbf{x}_0$, and we do LM:
>
> $$ p(\mathbf{x}\_2, \mathbf{x}\_1, \mathbf{x}\_0) =
> \underbrace{p(\mathbf{x}\_2 \mid \mathbf{x}\_1, \mathbf{x}\_0)}\_{\text{step 2}}
> \underbrace{p(\mathbf{x}\_1 \mid \mathbf{x}\_0)}\_{\text{step 1}}
> p(\mathbf{x}\_0) =
> \underbrace{\frac{p(\mathbf{x}\_2, \mathbf{x}\_1, \mathbf{x}\_0)}{p(\mathbf{x}\_1, \mathbf{x}\_0)}}\_{\text{step 2}}
> \underbrace{\frac{p(\mathbf{x}\_1, \mathbf{x}\_0)}{p(\mathbf{x}\_0)}}\_{\text{step 1}}
> p(\mathbf{x}\_0).$$
>
> If you see, by decomposition, the LM object as conditional probabilities implemented with softmax, and for the LM objective to be **correct,** it has to be that across time step 2 and time step 1, then:
>
> - $p(\mathbf{x}_1, \mathbf{x}_0)$ at the denominator in step 2
> - $p(\mathbf{x}_1, \mathbf{x}_0)$ at the numerator in step 1
>
> **must be equal.** If they are not equal, it means that the approach is not modeling correctly the probabilities of a sentence, probably producing content that is not coherent with the training data, out-of-distribution, or the model being more creative, thereby we could use the discrepancy between those two probabilities in the form of energies as a signal for hallucination detection.
>
> Also, please see the long response to the reviewer `qt4T` for **(W2, Q1)** for a detailed response on how we end up with:
>
> $$E(\mathbf{x}\_2,\mathbf{x}\_1,\mathbf{x}\_0) = E^l(\mathbf{x}\_2,\mathbf{x}\_1,\mathbf{x}\_0)\ \underbrace{- E^m(\mathbf{x}\_1,\mathbf{x}\_0) + E^l(\mathbf{x}\_1,\mathbf{x}0)}\_{\text{should be zero}}\ \underbrace{- E^m(\mathbf{x}\_0) + E^l(\mathbf{x}0)}\_{\text{should be zero}}$$
>
> We hope that this clarifies the reviewer's confusion and will increase the evaluation of our paper. We genuinely believe that our method can be a serious game-changer in LLM hallucination detection and believe that we have identified a crucial missing piece that may aid future research, not only in hallucination detection but also in better decoding with reasoning capabilities like [D].
>
>  Therefore, we hope the reviewer can increase the score.
>
> *Liu et al. “Energy-based out-of-distribution detection”. In NeurIPS, 2020.*
>
> [D] Karan, Aayush, and Yilun Du. "Reasoning with Sampling: Your Base Model is Smarter Than You Think." arXiv preprint arXiv:2510.14901 (2025).

---

> ### Author Response · Authors · 2025-11-23
> **Answer to Reviewer zjL6 (5/5)**
>
> ### (W5) Latency
>
> > Each time, we need to inference the model twice, the extra latency make the method impractical
>
> All current methods for hallucination detection must strike a balance between **their performance and the additional overhead** required to detect them. Unless we build machines that are natively capable of self-calibration, these types of methods may incur some overhead, as some researchers argue that self-calibration in LLMs is nearly impossible [A].
>
> We believe the limitation is not significant when compared to the limitations of other methods. Instead, selecting the **exact token is probably one of the mildest overheads across others**, and can be optimized for practical deployment.
>
> Please see below the **overheads and limitations of current methods**:
>
> | Method | Overhead & Limitations | Training Free |
> | --- | --- | --- |
> | Semantic Entropy | **Over Sampling (x10)** | **Yes** |
> | -- | Extra weights | Yes |
> |  |  |  |
> | Task-Dependent Classifier | Run the classifier  extraction | No |
> | -- | Overfit | No |
> | -- | Extra weights | No |
> | -- | Exact token | No |
> |  |  |  |
> | Spilled Energy (**Ours**) | Exact token extraction | **Yes** |
> |  |  |  |
>
> In our implementation, the exact answer tokens are detected by prompting a medium-sized LLM with the task of extracting a brief answer from the generated text in the original model answer. The brief answer is then located within the generated text, and the logits for the corresponding tokens are extracted for energy computations.
>
> Semantic Entropy is based on building clusters over generated answers via oversampling, which is basically the same as returning the same response to the answer at least x10 times, and also requires semantic clustering with a separate model, which has higher computational complexity compared to a forward pass in an LLM to find the exact tokens. For other approaches, such as the Task-Dependent Classifier, there are additional overheads, including running the classifier and storing the weights. Additionally, given the issue of overfitting, they may require a classifier for each dataset, among other considerations.
>
> This task of localization can be in the future further optimized either using Small Language Models (SLMs) or training a small sequential classifier, not to detect hallucinations from activations, but from the vector of energies across the vocabulary, and relate them across timesteps: the supervision for this small recurrent classifier may be obtained by distilling the current method in order to consider the most informative tokens.
>
> [A] Zhu et al. *”On the Calibration of Large Language Models and Alignment”* EMNLP 2023

---

> ### Comment · Area_Chair_Mbgu · 2025-11-28
>
> Dear Reviewer, the discussion period is about to close. We kindly ask you to participate in the discussion or update your score based on the authors' rebuttal before the deadline. Thank you for your time and valuable contribution!

---

### Official Review · Reviewer_zjL6 · 2025-10-30

**Soundness:** 3
**Presentation:** 3
**Contribution:** 3
**Rating:** 6
**Confidence:** 3

**Summary:**

The paper proposes a training-free method to detect hallucinations in Large Language Models (LLMs) by reinterpreting the final softmax classifier as an Energy-based Model (EBM). This reinterpretation allows the authors to decompose the sequence-to-sequence probability chain into multiple interacting EBMs.

The core contribution is the introduction of "spilled energy", which measures the discrepancy between two energy quantities that should be mathematically equal across consecutive generation steps, but differ in the LLM's implementation due to the way marginal and joint probabilities are computed. The authors also propose a complementary metric, marginal energy, which is measurable at a single step.

The paper empirically demonstrates that high spilled energy strongly correlates with LLM hallucinations. Crucially, their method is training-free and shows strong cross-dataset generalization across state-of-the-art open-sourced LLMs (LLaMa-3, Mistral, Qwen-3) and nine benchmarks including synthetic and real-world datasets. It localizes the signal to the "exact answer tokens" and outperforms the logit confidence baseline and prior probe classifier approaches, especially in cross-dataset settings where probe classifiers fail to generalize.

**Strengths:**

1. The idea of reinterpreting the LLM's autoregressive sequence modeling as a chain of Energy-based Models (EBMs) is novel and very interesting.
2. The implementation of the hallucination detector is simple and training free, but achieve a good generalization across synthetic and real-world datsets.
3. The technical motivation and derivations are sound, and the method is well mathmatically-grounded.
4. The paper is well-written and the presentation is clear.

**Weaknesses:**

1. There might be some technical details make the detector hard to be applicable in the real-world tasks.
    - The paper emphasizes that localizing the signal to the "exact answer tokens" is essential. However, correctly to find the exact answer tokens might be non-trivial. In the paper, authors identify this span by prompting the LLM for a brief answer seems to be a bit fragile to me. This is essentially a dependence on a pre-processing step that relies on LLM's outputs, which may introduce noise or additional complexity.
   - Compared to logit confidence, spill energy requires calculation of the quantities using two-step predictions, which may introduce the additional cost. Marginal energy only needs the one-step calculation but the results for two energy forms are quite mixed.
    - The analysis focuses on the "exact answer" tokens which can moslty be applied to QA or classification tasks. How well would the Spilled Energy perform on detecting other types of hallucinations common in LLMs, such as self-contradictions or source-attributable error that might occur earlier in a long-form generated answer or reasoning trace, outside of the specific final answer tokens?

**Questions:**

1. Why do we observe the cross-dataset results that for non-aligned models (Mistral), marginal energy sometimes slightly outperforms spilled energy?

2. Is there any insight why marginal energy is a superior signal in the pre-trained model setting, or why instruction-tuning seems to amplify the margin for spilled energy? Maybe a more in-depth analysis into comparing two forms of energy is interesting for the pre-trained and instruction-tuned models.

Other questions are listed in the bulletpoints of weakness as the justification.

---

> ### Author Response · Authors · 2025-11-22
> **Answer to Reviewer zjL6 (1/3)**
>
> ### Strengths and merits of the paper
>
> Thank you for the thoughtful and constructive feedback. We are pleased that several important strengths of our work have been highlighted, many of which were independently emphasized by the other reviewers as well:
>
> - the **novelty and conceptual interest** of reinterpreting the softmax layer as an EBM and exposing a principled internal consistency signal (`zjL6`: “novel and very interesting”; `LpMe`: “efficient, simple, generic, easy to implement”);
> - The fact that our detector is **training-free, simple to implement, and broadly applicable**, a point strongly underscored by `zjL6` (“simple and training free”), and mentioned by `Xm2r` (“lightweight and applicable to most LLMs”), as well as noted positively by `qt4T`;
> - The strength of our **cross-dataset and cross-model generalization**, emphasized by `zjL6` (“strong generalization across synthetic and real-world datasets”), with consistent reinforcement from `Xm2r` (“validated across model families”) and `LpMe` (“strong performance across a wide range of models and tasks”);
> - The reviewers’ shared positive assessment of the **clarity and soundness** of the paper (`zjL6`, `qt4T`, `LpMe` ) . **All reviewers** describe the paper as well-written, clear, and mathematically grounded.
> - The appreciation of the **breadth and rigor of our empirical evaluation**, including both synthetic arithmetic stress tests and diverse real-world datasets, was recognized explicitly by `zjL6` (“empirically demonstrates strong correlation with hallucinations across nine benchmarks”), `Xm2r` (“results on both synthetic and real-world datasets validate generalization”), and `LpMe` (“synthetic setup is well-designed; real-world benchmarks are representative”).
>
> We appreciate your positive assessment, and we are encouraged by the strong alignment across reviewers regarding the conceptual novelty, clarity, generality, and empirical robustness of our approach.
>
> ---
>
> We now address the remaining remarks, with the hope that they will enhance the evaluation of our work.

---

> ### Author Response · Authors · 2025-11-23
> **Answer to Reviewer zjL6 (2/3)**
>
> ### (W1) Hard to apply in real-world tasks
>
> We point out that the exact answer is not our contribution but was already known in the paper by Orgad *et al*, published at the previous ICLR 2025. In this sense, we simply confirm the finding of Orgad *et al. *****However***,*** the point of our paper is to demonstrate that we can achieve outstanding generalization performance with a training-free method by reinterpreting the LM chain as EBMs.
>
> All current methods for hallucination detection must strike a balance between **their performance and the additional overhead** required to detect them. Unless we build machines that are natively capable of self-calibration, these types of methods may incur some overhead, as some researchers argue that self-calibration in LLMs is nearly impossible [A].
>
> We believe the limitation is not significant when compared to the limitations of other methods. Instead, selecting the **exact token is probably one of the mildest overheads across others**, and can be optimized for practical deployment.
>
> Please see below the **overheads and limitations of current methods**:
>
> | Method | Overhead & Limitations | Training Free |
> | --- | --- | --- |
> | Semantic Entropy | **Over Sampling (x10)** | **Yes** |
> | -- | Extra weights | Yes |
> |  |  |  |
> | Task-Dependent Classifier | Run the classifier  extraction | No |
> | -- | Overfit | No |
> | -- | Extra weights | No |
> | -- | Exact token | No |
> |  |  |  |
> | Spilled Energy (**Ours**) | Exact token extraction | **Yes** |
> |  |  |  |
>
> In our implementation, the exact answer tokens are detected by prompting a medium-sized LLM with the task of extracting a brief answer from the generated text in the original model answer. The brief answer is then located within the generated text, and the logits for the corresponding tokens are extracted for energy computations.
>
> Semantic Entropy is based on building clusters over generated answers via oversampling, which is basically the same as returning the same response to the answer at least x10 times, and also requires semantic clustering with a separate model, which has higher computational complexity compared to a forward pass in an LLM to find the exact tokens. For other approaches, such as the Task-Dependent Classifier, there are additional overheads, including running the classifier and storing the weights. Additionally, given the issue of overfitting, they may require a classifier for each dataset, among other considerations.
>
> This task of localization can be in the future further optimized either using Small Language Models (SLMs) or training a small sequential classifier, not to detect hallucinations from activations, but from the vector of energies across the vocabulary, and relate them across timesteps: the supervision for this small recurrent classifier may be obtained by distilling the current method in order to consider the most informative tokens.
>
> Still, we want to highlight that **exact token performance is stable and reliable:** As shown in Table 3 of Appendix A.2 in Orgad et al.[1], Answer extraction has a very high success rate, ranging from 0.95 (Llama3-Instruct-8B) to 0.99 (Mistral-7B). The extracted answer must then be located within the original generated answer, which acts as an additional safety mechanism. We achieved similar accuracy for answer localization, as we started from the code implementation of Orgad et al.
>
> We report in the following table, the extraction success rate across the full datasets using Mistral-Instruct-v0.2, which has also been included in section B.1 of the appendix:
>
> | Dataset | Success Rate (%) |
> | --- | --- |
> | TriviaQA | 90.29 |
> | HotpotQA | 87.37 |
> | Movies | 93.61 |
> | Math | 87.59 |
> | HotpotQA-WC | 92.38 |
>
> Note that some datasets have been excluded (IMDB, Winobias, Winogrande, MNLI), since they have a finite set of possible answers which can be used to easily locate the exact answer within the model’s generation.
>
> > Compared to logit confidence, spill energy requires calculation of the quantities using two-step predictions, which may introduce the additional cost. Marginal energy only needs the one-step calculation but the results for two energy forms are quite mixed.
>
> For clarity in this work, we do not aim to condition the decoding process to minimize spilled energy.  Our approach is simple:
>
> - LLM performs decoding at all time-steps and saves the necessary information, which is just two scalar energy values for each decoding step. The LLM is producing these values, regardless of whether we do or do not perform hallucination detection.
> - Then we can go back and check whatever step we want.
>
> **So there is no additional cost in reading already available information across two time steps, so both marginal and spilled energy have the same complexity.**
>
> [A] Karpowicz, MichaĹ. "On the Fundamental Impossibility of Hallucination Control in Large Language Models." *arXiv preprint arXiv:2506.06382* (2025).

---

> ### Author Response · Authors · 2025-11-23
> **Answer to Reviewer zjL6 (3/3)**
>
> ### (Q1, Q2) Why is marginal energy better sometimes? Signals in pre-trained vs instruct models
>
> > Why do we observe the cross-dataset results that for non-aligned models (Mistral), marginal energy sometimes slightly outperforms spilled energy?
>
> > Is there any insight why marginal energy is a superior signal in the pre-trained model setting, or why instruction-tuning seems to amplify the margin for spilled energy? Maybe a more in-depth analysis into comparing two forms of energy is interesting for the pre-trained and instruction-tuned models.
>
> To better address this question, we **have incorporated additional results for the rebuttal for LLaMa and a dedicated paragraph in section 5.2. Please see the updated Tab. 1:** this enables a more comprehensive analysis and coverage. We now show an ablation on the performance of pre-trained LLMs and their instruction-tuned counterparts across two different families.
>
> This fact *could* be related to what is already known in the literature [A] where:
>
> - Pretraining LLMs with a large number of parameters and longer training improves calibration.
> - Instruction Tuning deteriorates model calibration.
>
> Also, in other surveys [B, C], authors found that instruction tuning degrades classical confidence metrics. For example, [B] reports in their section 3.1.3:
>
> > Gekhman et al. [98] analyzed the training dynamics of incorporating new factual knowledge during the Supervised Fine-Tuning (SFT) process and found that LLMs struggle to acquire such new knowledge effectively. Most importantly, they discovered a correlation between the acquisition of new knowledge through SFT and increased hallucinations, suggesting that introducing new factual knowledge encourages LLMs to hallucinate.
>
> suggesting that SFT increases the probability of hallucinations. In our experiments we find that marginal energy and logit energy may degrade in quality when the instruction-tuned model is compared with its pretrained counterpart, e.g. for Mistral in Tab.1. We also find that spilled energy’s performance remains consistent, suggesting better robustness.
>
> To conclude, we also believe that our spilled energy may open new decoding strategies, such as performing decoding to minimize spilled energy, or be linked to current research that shows how proper decoding may enable reasoning in pretrained LLMs without RLHF, see [D].
>
> [A] Zhu et al. *”On the Calibration of Large Language Models and Alignment”* EMNLP 2023
> [B] Huang al. *"A survey on hallucination in large language models: Principles, taxonomy, challenges, and open questions."* ACM Transactions on Information Systems 43, no. 2 (2025)
> [C] Ho, Z., Liang, S. and Tao, D., 2025. Review of Hallucination Understanding in Large Language and Vision Models. *arXiv preprint arXiv:2510.00034*.
> [D] Karan, Aayush, and Yilun Du. "Reasoning with Sampling: Your Base Model is Smarter Than You Think." arXiv preprint arXiv:2510.14901 (2025).

---

> ### Comment · Area_Chair_Mbgu · 2025-11-28
>
> Dear Reviewer, the discussion period is about to close. We kindly ask you to participate in the discussion or update your score based on the authors' rebuttal before the deadline. Thank you for your time and valuable contribution!

---

### Official Review · Reviewer_qt4T · 2025-10-30

**Soundness:** 2
**Presentation:** 3
**Contribution:** 2
**Rating:** 2
**Confidence:** 4

**Summary:**

This paper reinterprets the final softmax layer of large language models (LLMs) as an energy-based model (EBM). From this perspective, the authors claim to identify a discrepancy between two energy formulations, termed \emph{spilled energy}, which correlates strongly with hallucinations. Unlike probing classifiers, spilled energy requires no additional training. Experiments on LLaMa-3, Mistral, and Qwen-3 across nine benchmarks show that spilled energy outperforms logits and probing methods in detecting hallucinations in the cross dataset setup.

**Strengths:**

- The paper considers a very timely and important problem of hallucination detection.
- The paper is well-written and easy to follow.

**Weaknesses:**

- In my view, the reliance on exact tokens is a significant limitation. I am familiar with the paper by Orgad et al., which was the first to demonstrate that identifying the right token can improve detection. However, I see this primarily as an interesting observation rather than a practical method for hallucination detection. The reason is that identifying such tokens typically requires external algorithms or the use of other LLMs, which leads to very high latency—making it impractical for real-world hallucination detection.
- I find the concept of “spilled energy” somewhat unclear and potentially inconsistent. Could you please see my first question below for further context?
- The work appears to lack some important baselines, and the experimantal setup is unclear. For example, in my view, it would be valuable to include comparisons with Ptrue and Semantic Entropy [1]. In addition, some key comparisons are not clearly presented—for instance, I did not see results benchamarked against the baselines (even those already considered in the paper) under the standard setup (e.g., Orgad et al. when trained and tested on the same dataset). I would appreciate clarification on this point. If such a comparison is not included (and only the “cross-dataset” case is considered), it would significantly weaken the strength of the paper.

**References:**

[1] Kuhn et al., Semantic uncertainty: Linguistic invariances for uncertainty estimation in natural language generation, 2023

**Questions:**

- I have a question regarding the following sentence (quoting from the paper):
"E^{\ell}_{\theta} and E^{m}_{\theta} are computed from the output of the model, but with two key differences: E^{\ell}_{\theta} is obtained as a single logit extracted using the id of the sampled token, while E^{m}_{\theta} is computed by marginalizing over all id's in the vocabulary."
 It seems that you are associating the logit of a single token with the energy function of all the preceding tokens as well. Could you clarify why this is considered correct? This association feels inconsistent: if E^{\ell}_{\theta} corresponds to a single token, then why does E^{m}_{\theta} correspond to the normalization factor over the entire vocabulary? More concretely, I would have expected E^{m}_{\theta} to represent the prediction for the specific token immediately preceding the one associated with E^{\ell}_{\theta}, rather than the marginalized quantity. Consequently, the idea of “spilled energy” seems problematic, as it appears to rely on an inconsistency in how energy is defined. I would be happy to get a clarification on that.
- Could you clarify what is meant by “cross-dataset” in Table 1? This is not explained anywhere, and I find it difficult to understand the setup. Since your method does not require training, it is not clear to me how cross-dataset evaluation is applicable here. I do see how this makes sense in the context of Orgad et al.—since their probing method requires training—but then, what would be considered the standard experimental setup for them without cross-dataset evaluation (if its not Table 1)? Additionally, could you clarify how the cross-dataset setup in Table 1 is applied to Orgad et al.? Specifically, what is the exact setup—what was the model trained on, and what was it tested on?
- Could you clarify what the superscripts m and l represent in Equation (4)?
- In the Abstract, you write: “we propose a method to detect hallucinations completely training-free that naturally generalizes across tasks and LLMs.” Could you please point out where in the paper you demonstrate generalization across LLMs?

---

> ### Author Response · Authors · 2025-11-22
> **Answer to Reviewer qt4T (1/6)**
>
> ### Strengths and merits of the paper
>
> Thank you for the constructive feedback. We are pleased that Reviewer `qt4T` highlighted several important strengths of our work, many of which were independently emphasized by the other reviewers as well:
>
> - The **timeliness** and **relevance** of addressing hallucination detection in LLMs, a point echoed strongly by `zjL6` and `Xm2r`, who further note that our evaluation spans both synthetic and real-world settings representative of practical deployment;
> - the recognition that the paper is **well-written and easy to follow** (`qt4T`, `zjL6`, `LpMe`).
>
> ---
>
> We now address the remaining remarks, with the hope that they will enhance the evaluation of our work.
>
> ### (W1) Exact Token Limitation
>
> > In my view, the reliance on exact tokens is a significant limitation. I am familiar with the paper by Orgad et al., which was the first to demonstrate that identifying the right token can improve detection. However, I see this primarily as an interesting observation rather than a practical method for hallucination detection. The reason is that identifying such tokens typically requires external algorithms or the use of other LLMs, which leads to very high latency—making it impractical for real-world hallucination detection.
>
> All current methods for hallucination detection must strike a balance between **their performance and the additional overhead** required to detect them. Unless we build machines that are natively capable of self-calibration, these types of methods may incur some overhead, as some researchers argue that self-calibration in LLMs is nearly impossible [A].
>
> We believe the limitation is not significant when compared to the limitations of other methods. Instead, selecting the **exact token is probably one of the mildest overheads across others**, and can be optimized for practical deployment.
>
> Please see below the **overheads and limitations of current methods**:
>
> | Method | Overhead & Limitations | Training Free |
> | --- | --- | --- |
> | Semantic Entropy | **Over Sampling (x10 or more)** | **Yes** |
> | -- | Extra weights for embedder | Yes |
> |  |  |  |
> | Task-Dependent Classifier | Run the classifier | No |
> | -- | Overfit | No |
> | -- | Extra weights | No |
> | -- | Exact token | No |
> |  |  |  |
> | Spilled Energy (**Ours**) | Exact token extraction | **Yes** |
> |  |  |  |
>
> In our implementation, the exact answer tokens are detected by prompting a medium-sized LLM with the task of extracting a brief answer from the generated text in the original model answer. The brief answer is then located within the generated text, and the logits for the corresponding tokens are extracted for energy computations.
>
> Semantic Entropy is based on building clusters over generated answers via oversampling, which is basically the same as returning the same response to the answer at least x10 times, and also requires semantic clustering with a separate embedder model, which has higher computational complexity compared to a forward pass in an LLM to find the exact tokens. For other approaches, such as the Task-Dependent Classifier, there are additional overheads, including running the classifier and storing the weights. Additionally, given the issue of overfitting, they may require a classifier for each dataset, among other considerations.
>
> This task of localization can be in the future further optimized either using Small Language Models (SLMs) or training a small sequential classifier, not to detect hallucinations from activations, but from the vector of energies across the vocabulary, and relate them across timesteps: the supervision for this small recurrent classifier may be obtained by distilling the current method in order to consider the most informative tokens.
>
> Still, we want to highlight that **exact token performance is very stable and reliable:** As shown in Table 3 of Appendix A.2 in Orgad et al. (2025), Answer extraction has a very high success rate, ranging from 0.95 (Llama3-Instruct-8B) to 0.99 (Mistral-7B). The extracted answer must then be located within the original generated answer, which acts as an additional safety mechanism. We achieved similar accuracy for answer localization, as we started from the code implementation of Orgad et al.
>
> We report in the following table, the extraction success rate across the **full** datasets using Mistral-Instruct-v0.2, which has also been included in section B.1 of the appendix:
>
> | Dataset | Success Rate (%) |
> | --- | --- |
> | TriviaQA | 90.29 |
> | HotpotQA | 87.37 |
> | Movies | 93.61 |
> | Math | 87.59 |
> | HotpotQA-WC | 92.38 |
>
> Note that some datasets have been excluded (IMDB, Winobias, Winogrande, MNLI), since they have a finite set of possible answers which can be used to easily locate the exact answer within the model’s generation.
>
> [A] Karpowicz, MichaĹ. "On the Fundamental Impossibility of Hallucination Control in Large Language Models." *arXiv preprint arXiv:2506.06382* (2025).

---

> ### Author Response · Authors · 2025-11-22
> **Answer to Reviewer qt4T (2/6)**
>
> ### (W2, Q1) Spilled Energy is not clear and potentially inconsistent
>
> > W2) I find the concept of “spilled energy” somewhat unclear and potentially inconsistent. Could you please see my first question below for further context?
>
> We strive to make our paper and method as clear as possible. Please see the response below, which we hope will help clarify any points.
>
> > Q1) I have a question regarding the following sentence (quoting from the paper):” $E^{\ell}\_{\theta}$  and $E^{m}\_{\theta}$ are computed from the output of the model, but with two key differences: $E^{\ell}\_{\theta}$ is obtained as a single logit extracted using the id of the sampled token, while $E^{m}\_{\theta}$ is computed by marginalizing over all id's in the vocabulary." It seems that you are associating the logit of a single token with the energy function of all the preceding tokens as well. Could you clarify why this is considered correct?
>
> Yes, the reviewer's interpretation is correct. We additionally relate energies across two different time steps, $i$ and $i-1$, linking the logit value at time step $i$ with the marginalization across all vocabulary at time step $i-1$. See below in the next answer for our derivation and explanation.
>
> > This association feels inconsistent: if $E^{\ell}\_{\theta}$ corresponds to a single token, then why does $E^{m}\_{\theta}$ correspond to the normalization factor over the entire vocabulary? More concretely, I would have expected $E^{m}\_{\theta}$ to represent the prediction for the specific token immediately preceding the one associated with $E^{\ell}\_{\theta}$ , rather than the marginalized quantity. Consequently, the idea of “spilled energy” seems problematic, as it appears to rely on an inconsistency in how energy is defined. I would be happy to get a clarification on that.
>
> **TL;DR** Consider Eq. (2) in our paper and the simplification that occurs between the two probabilities between step $i$ and step $i-1$: that simplification occurs **because the probability in the denominator at step $i$ is the same as the probability in the numerator at step $i-1$ in order to perform language modeling correctly.** We measure those discrepancies inside LLMs in terms of energy, and the **spilled energy is the amount by which they differ.**
>
> Please see below for a longer and detailed response.

---

> ### Author Response · Authors · 2025-11-22
> **Answer to Reviewer qt4T (3/6)**
>
> Please see the definition below. Let us assume a sequence of three tokens  $\mathbf{x}_2, \mathbf{x}_1, \mathbf{x}_0$ . If we do language modeling with autoregression, minimizing the negative log-likelihood, we have:
>
> $$
> -\log p(\mathbf{x}\_2, \mathbf{x}\_1, \mathbf{x}\_0) = -\log \underbrace{p(\mathbf{x}\_2|\mathbf{x}\_1,\mathbf{x}_0)}\_{\text{step 2}}p(\mathbf{x}\_1|\mathbf{x}\_0)p(\mathbf{x}\_0)
> $$
>
> Now, every conditional probability on the right side is implemented with a transformer ending in a softmax discriminative classifier. Equations (3) and (5) in our paper allow us to re-interpret:
>
> $$
> \text{\textbf{step 2:}}\quad -\log p(\mathbf{x}_2|\mathbf{x}_1,\mathbf{x}_0) = -\log \frac{p(\mathbf{x}_2,\mathbf{x}_1,\mathbf{x}_0)}{p(\mathbf{x}_1,\mathbf{x}_0)} = -\log \Bigg[ \frac{ \exp\big(\theta(\mathbf{x}_1,\mathbf{x}_0)[id(\mathbf{x}_2)]\big)}{\sum_k^V\exp\big(\theta(\mathbf{x}_1,\mathbf{x}_0)[k]\big)} \Bigg] = E^l(\mathbf{x}_2,\mathbf{x}_1,\mathbf{x}_0)-E^m(\mathbf{x}_1,\mathbf{x}_0).
> $$
>
> In other words, we reinterpret:
>
> - the numerator $p(\mathbf{x}_2,\mathbf{x}_1,\mathbf{x}_0)$ as the energy $E^{\ell}(\mathbf{x}_2,\mathbf{x}_1,\mathbf{x}_0)$, which is the **logit ($\ell$)** of the softmax at timestep 2;
> - The denominator as the energy $E^m(\mathbf{x}_1,\mathbf{x}_0)$ obtained with the **marginalization ($m$)** across the vocabulary $V$. This value can be read  “read” simply by taking the denominator of the softmax at timestep 2. Please remember this term.
>
> It is better to indicate them as energies (since they are not probabilities), and given their logarithmic properties, we obtain a difference. We use the notation $l$ for logits and $m$ for marginalization.
>
> Now, **when we go across steps and we connect two-time steps, this is where the magic happens:**
>
> $\text{\textbf{step 1:}}\quad-\log p(\mathbf{x}_1|\mathbf{x}_0) = -\log \frac{p(\mathbf{x}_1,\mathbf{x}_0)}{p(\mathbf{x}_0)} = E^l(\mathbf{x}_1,\mathbf{x}_0)-E^m(\mathbf{x}_0).$
>
> We see that at timestep 1, the value $E^l(\mathbf{x}_1,\mathbf{x}_0)$ **appears again, but measured at the logit level.**
>
> In other words, across the time-steps 2 and 1, the quantity $E(\mathbf{x}_1,\mathbf{x}_0)$ is measured twice:
>
> - at timestep 2, as the marginalization
> - at timestep 1, as the logit.
>
> In the architecture or in the loss, there is no mechanism that is forcing this to be the **same, but they should be equal, given the language modeling objective.** This is the same as saying that in Equation (2) of our paper, the probabilities across time steps need to be simplified as we indicate.
>
> In other words, this:
>
> $p(\mathbf{x}_2, \mathbf{x}_1, \mathbf{x}_0) = p(\mathbf{x}_2|\mathbf{x}_1,\mathbf{x}_0)p(\mathbf{x}_1|\mathbf{x}_0)p(\mathbf{x}_0)$
>
> implies:
>
> $$E(\mathbf{x}\_2,\mathbf{x}\_1,\mathbf{x}\_0) = E^l(\mathbf{x}\_2,\mathbf{x}\_1,\mathbf{x}_0) \underbrace{- E^m(\mathbf{x}\_1,\mathbf{x}\_0) + E^l(\mathbf{x}\_1,\mathbf{x}\_0)}\_{\text{should be zero}} \underbrace{- E^m(\mathbf{x}\_0) + E^l(\mathbf{x}\_0)}\_{\text{should be zero}}$$
>
> To model the energy of a sequence $E^l(\mathbf{x}_2,\mathbf{x}_1,\mathbf{x}_0)$ correctly, then:
>
> - $- E^m(\mathbf{x}_1,\mathbf{x}_0) + E^l(\mathbf{x}_1,\mathbf{x}_0) = 0$ (spilled energy at timestep 2 if non-zero)
> - $-E^m(\mathbf{x}_0) + E^l(\mathbf{x}_0) = 0$ (spilled energy at timestep 1 if non-zero)
>
> so that $E(\mathbf{x}_2,\mathbf{x}_1,\mathbf{x}_0) = E^l(\mathbf{x}_2,\mathbf{x}_1,\mathbf{x}_0)$.
>
> > I would have expected $E^{m}\_{\theta}$ to represent the prediction for the specific token immediately preceding the one associated with $E^{\ell}_{\theta}$ , rather than the marginalized quantity.
>
> If we define the spilled energy in the suggested way, there is no rationale that supports this definition; in fact, we would be measuring the discrepancy between $E^l(\mathbf{x}_2,\mathbf{x}_1,\mathbf{x}_0)$  and $E^l(\mathbf{x}_1,\mathbf{x}_0)$ that:
>
> - relates to energies to different input tokens $(\mathbf{x}_2,\mathbf{x}_1,\mathbf{x}_0)$  vs  $(\mathbf{x}_1,\mathbf{x}_0)$
> - These two quantities are not intended to be equal according to the language modeling objective.
>
> In light of this, we have demonstrated that **our method is consistent** and **guided by a clear rationale.** We hope that this brief proof will help clarify any misunderstandings.

---

> ### Author Response · Authors · 2025-11-22
> **Answer to Reviewer qt4T (4/6)**
>
> ### (Q3) Clarification on superscripts $m$ and $\ell$
>
> > Could you clarify what the superscripts m and l represent in Equation (4)?
>
> If it was not clear from the previous explanation, $\ell$ means we are measuring the energy at the **logit** level of the current token (i.e., numerator of softmax) while $m$ means the **marginalized** level, thus taking it from the denominator of softmax, so:
>
> - $\ell$ as logit (numerator of softmax)
> - $m$ as marginalized (denominator of softmax)
>
> ---
>
> ### (Q4) Clarification on Generalization of Spilled Energy across LLMs
>
> > In the Abstract, you write: “we propose a method to detect hallucinations completely training-free that naturally generalizes across tasks and LLMs.” Could you please point out where in the paper you demonstrate generalization across LLMs?
>
> Given that our approach is training-free, i.e., we do not train on a specific LLM and then test on another LLM, we can apply our method arbitrarily to *any LLM or sequential model provided with softmax and cross-entropy loss for language modeling.*
>
> The generalization across LLMs is shown with strong evidence in **Table 1 and Fig. 3,**, we also added new evidence in Tab. 3. So far we tested the following different LLMs:
>
> - LLaMA3-Instruct (Tab. 1)
> - LLaMA3 (Tab. 1)
> - Mistral (Tab. 1)
> - Mistral-Instruct (Tab. 1)
> - Gemma-Instruct (Tab. 3)
> - Qwen-3 8B (Fig. 3)
>
> **Our method generalizes well across all these different LLMs, whether pretrained or instruction-tuned. In Table 2, across 4 LLMs, we have an average AUC across 9 datasets of 73.32, much higher than the other baselines.**

---

> ### Author Response · Authors · 2025-11-22
> **Answer to Reviewer qt4T (5/6)**
>
> ### (W3) Clarification on Baselines and Evaluation Setup
>
> **Inclusion of P(True) and Semantic Entropy**
>
> We have included **P(True)** as a key baseline in our evaluation, and the results are presented in **Table 1** of the revised paper.
>
> Regarding **Semantic Entropy (SE)**, we chose not to include it for the following two primary reasons:
>
> 1. **Computational Intractability:** The original SE approach requires generating and analyzing multiple samples per query, typically **5 to 10** or up to **20**, to perform semantic clustering. Given our extensive evaluation across $\sim 9$ diverse datasets and $4$ state-of-the-art LLMs, this would introduce a prohibitive 5-10x increase in computational cost for what is already a large-scale analysis. Methods like Semantic Entropy Probes (SEPs) [B] address this by approximating SE from a single forward pass, but this requires training a probe, which would be captured in the spirit of the existing trained probe baselines, and their generalization issues.
> 2. **Unclear Methodology for Long-Form Generation:** While SE works well for short-form generations, its application to **long reasoning traces** or free-form long answers is complex and not fully defined or consistently implemented in publicly available code, requiring decomposition into factoids. Since our analysis also includes HotpotQA without context and the Math datasets, we wouldn't be able to ensure a fair comparison.
>
> [B] Kossen, J., Han, J., Razzak, M., Schut, L., Malik, S.A., & Gal, Y. (2024). Semantic Entropy Probes: Robust and Cheap Hallucination Detection in LLMs. ArXiv, abs/2406.15927.

---

> ### Author Response · Authors · 2025-11-22
> **Answer to Reviewer qt4T (6/6)**
>
> **Cross-Dataset vs. In-Distribution Probes**
>
> The core strength of our method is its training-free nature and **robust cross-dataset generalization**, making it far more practical for real-world deployment than probes, which require training data for every task.
>
> While the paper explicitly focuses on the cross-dataset scenario in the main text (Table 1), we have provided the comprehensive performance matrix—including the **in-distribution** case—for the Orgad et al. probes **in the new Fig. 4 in the main paper and in additional Figs. 8, 9, 10 in the Appendix.**
>
> This comparison demonstrates how poorly the probes generalize to different tasks, thus validating our choice to pursue a training-free approach that generalizes immediately.
>
> Fig.4 in the paper is a “confusion” matrix of the probe performance within and across datasets, along with a comparison with our spilled energy method. This matrix, shown as a heatmap,  captures all in-distribution (diagonal elements) and out-of-distribution performance ( off-diagonal elements). We present the tables below, along with our discussion of the results.
>
> | Train\Test |   TriviaQA |   HotpotQA |   Movies |   Winobias |   Winogrande |   MNLI |   IMDB |   Math |   HotpotQA-WC |
> |:------------|-----------:|-----------:|---------:|-----------:|-------------:|-------:|-------:|-------:|--------------:|
> | TriviaQA |         84 |         74 |       71 |         74 |           55 |     59 |     56 |     83 |            59 |
> | HotpotQA |         78 |         83 |       74 |         53 |           59 |     55 |     51 |     72 |            64 |
> | Movies |         69 |         69 |       82 |         72 |           55 |     52 |     72 |     52 |            52 |
> | Winobias |         57 |         55 |       61 |         93 |           52 |     53 |     70 |     51 |            52 |
> | Winogrande |         54 |         56 |       67 |         63 |           78 |     69 |     81 |     50 |            52 |
> | MNLI |         55 |         63 |       61 |         63 |           57 |     91 |     81 |     59 |            52 |
> | IMDB |         55 |         60 |       65 |         70 |           57 |     55 |     96 |     54 |            61 |
> | Math |         58 |         67 |       56 |         63 |           53 |     58 |     54 |     95 |            63 |
> | HotpotQA-WC |         59 |         72 |       61 |         55 |           56 |     53 |     67 |     83 |            76 |
>
> AuROC performance of the method proposed by Orgad et al. (2025) using Llama-Instruct 8B.
>
> We also present a comparison with our Spilled Energy method performance:
>
> | Train\Test | TriviaQA | HotpotQA | Movies | Winobias | Winogrande | MNLI | IMDB | Math | HotpotQA-WC |
> | --- | --- | --- | --- | --- | --- | --- | --- | --- | --- |
> | TriviaQA | **3.07** | **9.07** | **18.07** | **30.07** | **33.07** | **32.07** | **32.07** | **29.07** | **28.07** |
> | HotpotQA | **11.98** | **2.98** | **16.98** | **30.98** | **29.98** | **22.98** | **25.98** | **18.98** | **13.98** |
> | Movies | **18.34** | **15.34** | **7.34** | **28.34** | **22.34** | **28.34** | **24.34** | **33.34** | **28.34** |
> | Winobias | -13.28 | **7.72** | -11.28 | -32.28 | -2.28 | -2.28 | -9.28 | -2.28 | **5.72** |
> | Winogrande | **0.11** | -3.89 | **0.11** | **3.11** | -22.89 | -1.89 | -1.89 | **2.11** | -0.89 |
> | MNLI | **14.95** | **18.95** | **21.95** | **20.95** | **4.95** | -17.05 | **18.95** | **15.95** | **20.95** |
> | IMDB | -8.34 | -3.34 | -24.34 | -22.34 | -33.34 | -33.34 | -48.34 | -6.34 | -19.34 |
> | Math | -17.42 | -6.42 | **13.58** | **14.58** | **15.58** | **6.58** | **11.58** | -29.42 | -17.42 |
> | HotpotQA-WC | **34.00** | **29.00** | **41.00** | **41.00** | **41.00** | **41.00** | **32.00** | **30.00** | **17.00** |
> | Average | **4.82** | **7.71** | **9.27** | **12.71** | **9.82** | **8.49** | **9.49** | **10.16** | **8.49** |
>
> AuROC Performance Matrix Comparison (Llama-3-Instruct): Difference for In-Distribution performance is on the diagonal, and difference for Cross-Dataset performance is on the off-diagona. This table depicts the performance difference between their method and our Spilled $\Delta E$ with Min pooling. Positive values indicate cases where Spilled $\Delta E$ outperforms probing classifiers. All the numbers are expressed as percentages. The diagonal elements (in bold) confirm the high performance of the probes when evaluated in-distribution, but the sharp drop observed in the off-diagonal elements supports our premise that this standard, in-distribution setup significantly overestimates the utility of trained probes for broad LLM deployment.

---

> > ### Author Response · Authors · 2025-11-22
> >
> > We firmly believe that our method can be a genuine game-changer in LLM hallucination detection and believe that we have identified a crucial missing piece that may aid future research. Therefore, we hope the reviewer can increase the score, given the current very low score.

---

> > > ### Comment · Reviewer_qt4T · 2025-11-27
> > > **Response to authors**
> > >
> > > I thank the authors for their detailed rebuttal, I will respond to each of their points.
> > >
> > > ---
> > >
> > > *Exact Token Limitation*
> > >
> > > I have read the authors’ response, but I still view exact-token matching as a limitation -- particularly because identifying signals localized to specific tokens has already been explored in prior work, such as Orgad et al. In that paper, the finding that the signal is localized to specific tokens was itself a key contribution, rather than merely a mechanism for hallucination detection.
> > >
> > > The authors attempt to justify the practicality of their method by contrasting it with classifiers, but this comparison is not very compelling. Classifiers can typically be trained with something as lightweight as logistic regression, which is extremely fast, and their inference speed is also extremely high.
> > >
> > > ---
> > >
> > > *Spilled Energy is Not Clear and Potentially Inconsistent*
> > >
> > > I carefully read the reviewers’ explanation, but the concept still feels unclear. The authors appear to assume – intuitively – that the two quantities $E^m$ and $E^l$ should coincide.
> > >  Other reviewers also expressed similar confusion. As it stands, the method seems to rely on a heuristic that works empirically, but the limited evaluation and relatively high latency weaken the overall contribution.
> > >
> > > ---
> > >
> > > *Clarification on Superscripts*
> > >
> > > Thank you for the clarification; I now understand this point.
> > >
> > > ---
> > >
> > > *Clarification on Generalization of Spilled Energy across LLMs, Cross-Dataset vs. In-Distribution Probes*
> > >
> > > Thank you for the explanation. In most cases, “generalization across LLMs/tasks” refers to training on one and testing on another. Since no actual training is involved here, this naming initially caused confusion for me, but your explanation clarified the intent.
> > >
> > > ---
> > >
> > > *Clarification on Baselines and Evaluation Setup*
> > >
> > > I still find the response insufficient.
> > > The scope of the baselines used in this paper is important because I see the paper’s primary contribution as a hallucination detector. Given that the method is relatively slow (as noted above) and appears (to me) to rely on a heuristic rather than a clearly grounded principle, the quality and breadth of the empirical evaluation become especially important. Without stronger evaluation, it is difficult to assess the true effectiveness of the approach.
> > >
> > > ---
> > >
> > > Overall, my main concern remains that the definition of Spilled Energy is still not clearly articulated, and other reviewers expressed similar confusion. The approach appears more like a workable heuristic than a well-founded metric, and the limited empirical evaluation does not provide strong evidence of its robustness. Moreover, the method relies on an additional language model, which undermines the claim of being a real-time detection technique – a concern raised by other reviewers as well.

---

> ### Author Response · Authors · 2025-11-28
> **Addressing Inconsistencies regarding Derivation, Scope, and Empirical Rigor (1/2)**
>
> We thank the reviewer for their comments. However, we must respectfully correct several **factual misunderstandings** regarding our derivation, the computational reality of the baselines, and the true scope of our contribution. We address these point-by-point below.
>
> ---
>
> ## On the "Inconsistency" of Spilled Energy and Peer Consensus
>
> The reviewer states:
>
> > "The concept still feels unclear... As it stands, the method seems to rely on a heuristic... other reviewers expressed similar confusion."
>
> **This conclusion, based on peer consensus, is factually incorrect.** We point to the actual text of the other reviews, which explicitly praise the mathematical clarity and soundness of the derivation:
>
> - **Reviewer** `zjL6`:
> >  "The technical motivation and derivations are **sound**, and the method is **well mathematically-grounded**."
> - **Reviewer** `LpMe`:
> >  "The entire derivation of spilled energy is presented in the main text... The derivation is **clear and easy to follow**."
>
> Regarding the "heuristic" claim: Our method is not a heuristic; it is the direct application of the **Chain Rule of Probability** $P(A,B) = P(A|B)P(B)$ mapped to the EBM framework in a **sequence modeling** setting. The "spilled energy" is the measurement of the violation of this probability chain rule in terms of energy, ***not merely an intuition or a heuristic.***
>
> If the reviewer believes this derivation is "inconsistent," we respectfully ask them to **point out exactly which step of the derivation (Eq.s 1-8) is mathematically flawed**, rather than stating an opinion that contradicts the established consensus of the other reviewers.
>
> ---
>
> ## On "Fast" Classifiers and Generalization
>
> The reviewer argues:
>
> > "Classifiers can typically be trained with something as lightweight as logistic regression, which is extremely fast, and their inference speed is also extremely high. This comparison is not very compelling."
>
> This argument overlooks several critical facts:
>
> 1. **Training Cost:** While the logistic regression training update is fast, **data collection is expensive**. Training a probe requires running the full LLM forward pass on thousands of examples to collect hidden states across layers. Our method requires **zero training** and **zero collection of hidden states for training.**
> 2. **Classifiers need supervision, whereas Ours is unsupervised.** To train a classifier, it requires supervision from a labeled dataset; spilled energy is generic, so we do not need to train, and thus it is unsupervised.
> 3. **Which Token and Activation?** Even for classifiers, you have to select on which token to compute the activation, and also at which layer you select the activation. Is the activation selection stable across different LLMs for classifiers? **Our method does not require any activation selection and is stable across different LLMs, as proven in Tab. 1.**
> 4. **On which data to train the classifier?** Think about delivering the detection system into a general-purpose LLM, on which data the classifier needs to be trained to generalize well? Our new Fig. 4(a) confirms that on cross-testing, classifiers do not generalize well, and Fig.4(b): our method greatly improves the off-diagonal. **These are facts and evidence that support our method.**
> 5. **Generalization Failure**: The reviewer suggests classifiers are a viable alternative, yet *Orgad et al.* themselves **explicitly** state:
>
> > We show that such error detectors **fail to generalize** across datasets, implying that—contrary to prior claims—truthfulness encoding is not universal but rather multifaceted.
>
> Furthermore, the reviewer criticizes our use of "exact tokens" as a limitation, yet **probing classifiers** *also* **require exact token localization** to achieve their reported performance, ***especially* in long generations**. It is **contradictory** to penalize our method for a **requirement that applies equally to the baseline the reviewer defends**.
>
> **Our method succeeds** precisely where classifiers *fail*: cross-dataset **generalization** and system overhead, including computational **cost**.

---

> ### Author Response · Authors · 2025-11-28
> **Addressing Inconsistencies regarding Derivation, Scope, and Empirical Rigor (2/2)**
>
> ## On Baselines and Latency
>
> The reviewer claims:
>
> > "The work appears to lack some important baselines... comparisons with P(True) and Semantic Entropy."
>
> - **P(True):** We **did** include $P(\text{True})$ in our evaluation (see **Table 1**). It is a computationally efficient baseline, but our method consistently outperforms it.
> - **Semantic Entropy (SE):** The reviewer criticizes our method for "**latency**" and the **reliance on other LLMs**, while simultaneously requesting SE. This is a **logical contradiction**.
>     - **Our Method:** Requires 1 generation pass, an extraction step (potentially with same model as the generation’s) and a very lightweight verification step.
>     - Semantic Entropy: Requires generating **x10 sample sequences per query and then clustering them with a separate model (DeBERTa).**
>
>     SE is ***inherently*** an order of magnitude computationally heavier. Furthermore, SE was originally designed for short-form QA. We follow the rigorous, realistic evaluation setting of Orgad et al. (2025), which involves long-form reasoning where SE is less applicable.
>
> ---
>
> ## On the Scope of the Contribution
>
> The reviewer states:
>
> > "I see the paper’s primary contribution as a hallucination detector."
>
> This perspective is reductive. While we evaluate our method as a detector to demonstrate empirical validity, our primary contribution is the **novel theoretical framework** of reinterpreting the LLM softmax layer as an Energy-Based Model (EBM). This introduces a new metric, **Spilled Energy**, which opens avenues far beyond post-hoc detection, including:
>
> - **Regularization:** Using $\Delta E$ as a penalty term during training to enforce internal consistency.
> - **Decoding Guidance:** Using $\Delta E$ to steer generation (e.g., EBM-guided decoding) toward more truthful outputs.
>
> This broader theoretical value is **recognized by the other reviewers:**
>
> - **Reviewer** `zjL6`:
>
> > "The idea of reinterpreting the LLM's autoregressive sequence modeling as a chain of Energy-based Models (EBMs) is **novel and very interesting**."
>
> - **Reviewer** `Xm2r`:
>
> > "The paper formulate the LLM's generation procedure as energy based model, which **enable the following definition of 'spilled energy'**."
>
> ---
>
> ## On "Limited Empirical Evaluation"
>
> The reviewer states:
>
> > "The limited empirical evaluation does not provide strong evidence."
>
>
> We strongly disagree with the characterization of our evaluation as "limited." We have evaluated our method on **3 Model Families** (Llama-3, Mistral, Gemma), **2 Training Regimes** (Instruction-Tuned and Pretrained), **9 Diverse Datasets**, and a **Synthetic Stress Test** (up to 13-digit sums).
>
> This characterization stands in **stark contrast to** the **explicit consensus of the other three reviewers**, who found the evaluation robust and comprehensive:
>
> - **Reviewer** `Xm2r`:
>
> > "The experiments in the paper are **solide**... The results on realworld datasets shows the method's valid on a **wide** range of domains. The results are validated **across model family** as well."
>
> - **Reviewer** `LpMe`:
>
> > "The proposed methods are evaluated on a **wide range of models and tasks**, and show **strong performance**... The synthetic setup is well-designed"
>
> - **Reviewer** `zjL6`:
>
> > "Achieve a **good generalization across synthetic and real-world datsets**... empirically demonstrates strong correlation with hallucinations across nine benchmarks"
>
> ---
>
> We believe we have presented a mathematically grounded, training-free metric that offers **both immediate utility** in hallucination detection **and broader theoretical potential** for LLM optimization. We hope these clarifications address the misunderstandings regarding our derivation’s theoretical grounding, baselines, and the *significance* of our contribution.

---

### Author Response · Authors · 2025-12-02
**Summary of Revisions, Reviewer Consensus, and Response to Outlier Concerns**

We thank the reviewers and the Area Chairs for the constructive discussion. Below, we summarize the strong consensus across the majority of reviewers, clarify the factual status of outstanding concerns, and outline the revisions that strengthen the paper’s contribution.

---

## **Broad Reviewer Agreement on Core Strengths**

Three reviewers (`zjL6`, `Xm2r`, `LpMe`) independently highlighted the same core strengths, offering a consistent and positive assessment of the work's value:

* **Novelty and Theoretical Value:**
    * `zjL6`: Describes the reinterpretation of LLMs as a chain of EBMs as **"novel and very interesting."**
    * `Xm2r`: Notes the formulation **"enables the definition of 'spilled energy'"** as a strength.
    * `LpMe`: States the derivation is **"clear and easy to follow."**
* **Practicality:**
    * `zjL6`: "Simple and training free."
    * `Xm2r`: "Lightweight and applicable to most LLMs."
    * `LpMe`: "Efficient, simple, generic, easy to implement."
* **Strong Experimental Evidence:**
    * `zjL6`: "Strong generalization across synthetic and real-world datasets."
    * `Xm2r`: "Validated across model family... robust across domains."
    * `LpMe`: "Evaluated on a wide range of models and tasks... strong performance."

**Regarding Scores:**
* **`LpMe` (Score 6):** Explicitly praises the efficiency and "strong performance" across the "wide range of models and tasks."
* **`zjL6` (Score 6):** Stated, *"Given a strong answer to Q1... I would be open to increasing the score."* We provided the requested derivation and additional analysis addressing this specific question.
* **`Xm2r` (Score 4):** Despite the conservative score, they explicitly note they **"would not mind if paper is accepted"** and assess the contribution, soundness, and presentation all as "3: Good."

## **Addressing Factual Inconsistencies in Reviewer qt4T’s Assessment**

We highlight that several of Reviewer `qt4T`'s (Score 2) objections are **factually unsupported** and **directly contradicted** by the other three experts.

* **Claim:** "Spilled Energy is unclear/inconsistent."
    * **Fact:** Contradicted by **all three** others. `zjL6` calls the motivation "sound" and "mathematically-grounded"; `LpMe` calls it "clear." We added **Appendix A.3** to explicitly link $\Delta E$ to the Chain Rule of Probability $P(A,B) = P(A|B)P(B)$, showing that Spilled Energy is the measurement of the violation of this rule in sequence modeling.
* **Claim:** "Exact token extraction is impractical."
    * **Fact:** Contradicted by prior work (Orgad et al., ICLR 2025) and our data showing **>90% success rates** across datasets. Furthermore, baseline classifiers *also* require token localization for optimal performance, making this a standard requirement for high-precision detection.
* **Claim:** "Limited evaluation."
    * **Fact:** Contradicted by the record. We evaluate **9 datasets**, **3 model families** (Llama, Mistral, Gemma), **2 training regimes** (Base vs. Instruct), and synthetic stress tests. Other reviewers explicitly praised this breadth.
* **Claim:** "Method causes latency." (while requesting Semantic Entropy)
    * **Fact:** A logical contradiction. The reviewer rejected our method based on latency but requested **Semantic Entropy**, a baseline requiring **10x more generation passes** per query followed by a separate clustering step with LLMs.

## **Summary of Revisions**

We updated the manuscript to include clarifications and further evidence:

* **Expanded Baselines (Table 1):** Added $P(\text{True})$ (addressing `qt4T`) and results for **Gemma** (Table 3) to demonstrate generalizability across architectures.
* **Pre-trained vs. Instruction-Tuned Analysis (Section 5.2):** Added analysis showing how classical uncertainty metrics degrade on instruction-tuned models (citing *Zhu et al.*), whereas Spilled Energy remains robust (addressing `zjL6`).
* **Cross-Dataset Generalization (Figure 4):** Added a "confusion matrix" heatmap contrasting the failure of trained probes in cross-dataset generalization (off-diagonal performance drops) against our robust performance (addressing `qt4T`, `LpMe`).
* **Transparency:** Added **Section A.3** (streamlined derivation from Probability Chain Rule) and **Section B.1** (exact token extraction details).

## **Impact on Future Research**

This work establishes a new theoretical lens for LLM reliability. By quantifying **probability chain rule** violations via Spilled Energy ($\Delta E$), we open avenues for:
* **Training:** $\Delta E$ as an auxiliary loss to enforce internal consistency during pre-training or SFT.
* **Decoding:** Constraints to dynamically reject hallucinated steps that exhibit high energy spills.
* **Efficiency:** Optimization via distillation into Small Language Models (SLMs) to detect errors in real-time.

The strong agreement across reviewers `zjL6`, `LpMe`, and `Xm2r`, combined with our factual clarifications, supports acceptance. We thank the AC for their consideration.

---

### Meta-Review · Area_Chair_e9Vn · 2026-01-07

**Summary:**

**Strengths**

1. Timely and important problem (hallucination detection)
(qt4T, zjL6, Xm2r, LpMe)

2. Novel energy-based interpretation of LLM autoregressive generation
(zjL6, Xm2r, LpMe)

3. Training-free, simple, and gray-box applicable (logits only)
(zjL6, Xm2r, LpMe)

4. Strong empirical results on synthetic and real-world datasets, with robustness across models/tasks
(zjL6, Xm2r, LpMe)

5. Clear writing and well-presented derivations
(qt4T, zjL6, LpMe)

**Weaknesses**

1. Strong reliance on accurately identifying “exact answer tokens,” which may be fragile or unreliable
(zjL6, Xm2r, LpMe, qt4T)

2. Limited applicability beyond short-answer or well-localized tasks (e.g., long-form or distributed hallucinations)
(zjL6, Xm2r)

3. Unclear or insufficient theoretical justification for why spilled energy indicates hallucinations
(qt4T, LpMe, Xm2r)

4. Additional inference cost and latency due to multi-step or dual-energy computation
(qt4T, zjL6, Xm2r)

5. Missing or unclear experimental baselines and evaluation setup (e.g., Semantic Entropy, P(true), cross-dataset definition)
(qt4T, LpMe)

6. Overstated or insufficiently supported claims about generalization across LLMs
(qt4T)

**Reviewer Concerns:**

**Addressed:**

- Limited applicability beyond short-answer or well-localized tasks (e.g., long-form or distributed hallucinations)
(zjL6, Xm2r)

- Additional inference cost and latency due to multi-step or dual-energy computation
(qt4T, zjL6, Xm2r)

- Missing or unclear experimental baselines and evaluation setup (e.g., Semantic Entropy, P(true), cross-dataset definition)
(qt4T, LpMe)

- Overstated or insufficiently supported claims about generalization across LLMs
(qt4T)

**Not Totally Resolved:**

-  Strong reliance on accurately identifying “exact answer tokens,” which may be fragile or unreliable
(zjL6, Xm2r, LpMe, qt4T)
- Unclear or insufficient theoretical justification for why spilled energy indicates hallucinations
(qt4T, LpMe, Xm2r)

**Reviewer Scores:**

- Reviewer qt4T: 2 -> 2
- Reviewer zjL6: 6 -> 6
- Reviewer Xm2r: 4 -> 6
- Reviewer LpMe: 6 -> 6

---

### Decision · Program_Chairs · 2026-01-26

Accept (Poster)